

# Genome-scale identification of *plant defensin* (*PDF*) family genes and molecular characterization of their responses to diverse nutrient stresses in allotetraploid rapeseed

Ying Liu[1,*], Ying-peng Hua[1,*], Huan Chen[2], Ting Zhou[1], Cai-peng Yue[1] and Jin-yong Huang[1]

[1] School of Agricultural Sciences, Zhengzhou University, Zhengzhou, China
[2] National Tobacco Quality Supervision and Inspection Center, Zhengzhou, China
* These authors contributed equally to this work.

Corresponding author
Jin-yong Huang,
jinyhuang@zzu.edu.cn

## ABSTRACT

Plant defensins (PDFs), short peptides with strong antibacterial activity, play important roles in plant growth, development, and stress resistance. However, there are few systematic analyses on *PDFs* in *Brassica napus*. Here, bioinformatics methods were used to identify genome-wide *PDFs* in *Brassica napus*, and systematically analyze physicochemical properties, expansion pattern, phylogeny, and expression profiling of *BnaPDFs* under diverse nutrient stresses. A total of 37 full-length *PDF* homologs, divided into two subgroups (*PDF1s* and *PDF2s*), were identified in the rapeseed genome. A total of two distinct clades were identified in the BnaPDF phylogeny. Clade specific conserved motifs were identified within each clade respectively. Most *BnaPDFs* were proved to undergo powerful purified selection. The *PDF* members had enriched *cis*-elements related to growth and development, hormone response, environmental stress response in their promoter regions. GO annotations indicate that the functional pathways of *BnaPDFs* are mainly involved in cells killing and plant defense responses. In addition, bna-miRNA164 and bna-miRNA172 respectively regulate the expression of their targets *BnaA2.PDF2.5* and *BnaC7.PDF2.6*. The expression patterns of *BnaPDFs* were analyzed in different tissues. *BnaPDF1.2bs* was mainly expressed in the roots, whereas *BnaPDF2.2s* and *BnaPDF2.3s* were both expressed in stamen, pericarp, silique, and stem. However, the other *BnaPDF* members showed low expression levels in various tissues. Differential expression of *BnaPDFs* under nitrate limitation, ammonium excess, phosphorus starvation, potassium deficiency, cadmium toxicity, and salt stress indicated that they might participate in different nutrient stress resistance. The genome-wide identification and characterization of *BnaPDFs* will enrich understanding of their molecular characteristics and provide elite gene resources for genetic improvement of rapeseed resistance to nutrient stresses.

## INTRODUCTION

Plant antimicrobial peptides (AMPs), short peptides produced by plants, are the first line of defense against bacterial and fungal invasion (*Ebbensgaard et al., 2015*). There are many types of AMPs, including plant defensins (PDFs), hevein-like peptides, knottin-type peptides, lipid transfer proteins, α-hairpinin, and snakins(*Nawrot et al., 2014*).

As an important part of the innate immune system of plants, *PDFs* are ubiquitously expressed in seeds, roots, stems, leaves, flowers, and fruits (*Silverstein et al., 2007*; *Parisi et al., 2019*). PDFs, first found in wheat endosperm, were originally classified as γ-thionine, which were later changed to the PDFs after being found similar in sequences and structures to insect defensins (*Mendez et al., 1990*; *Broekaert et al., 1995*). PDFs, containing 45–54 amino acids, are rich in cysteine (Cys) and basic amino acids (lysine and arginine). Eight cysteines form four pairs of disulfide bonds to stabilize α helix and three β folding sheets, forming CSαβ conformation (*Lay & Anderson, 2005*; *Sathoff et al., 2019*; *Odintsova, Slezina & Istomina, 2020*). Cys pairing patterns in PDFs are Cys1-Cys8, Cys2-Cys5, Cys3-Cys6, and Cys4-Cys7 (*García-Olmedo et al., 1998*). The characteristic sequences of α-helix and β-strand are Cys-Xaa-Xaa-Xaa-Cys and Cys-Xaa-Cys, respectively, both of which are common to most PDFs. Although the CSαβ motif of plant defensins is conserved, the amino acid sequences of their primary structure differ greatly (*Tam et al., 2015*).

Current studies have shown that PDFs play important roles in plant growth and development, and have strong antibacterial activity (*Van Loon, Rep & Pieterse, 2006*). Most PDFs display antibacterial activity mainly by changing membrane potential and permeability, destroying plasma membrane integrity, or inhibiting bacterial growth (*Tam et al., 2015*). The expression of *Raphanus sativus* defensin RsAFP2 in wheat enhanced the resistance to *Fusarium graminearum* and *Rhizoctonia cerealis* (*Li et al., 2011*). *Nicotiana alata* defensin NaD1 enters the cytoplasm of *Fusarium oxysporum* hyphae, resulting in granulation of the cytoplasm and cell death (*van der Weerden, Lay & Anderson, 2008*)

PDFs are also inhibitors of trypsin and amylase, and they can block ion channels and enhance metal tolerance. VrD1 and TvD1 exhibit insecticidal activity against alpha-amylase (*Lin et al., 2007*; *Vijayan et al., 2012*). It has been proved that Arg at position 38 of MsDef1 is key to antifungal activity. In addition, *MsDef1* also effectively blocks $Ca^{2+}$ channels (*Spelbrink et al., 2004*). The *Arabidopsis thaliana* defensin AtPDF2.3 contains a toxin signature sequence (K-C5-R-G), which effectively blocks potassium channels (*Vriens et al., 2016*). PDFs are also involved in heavy metal detoxification. Overexpressed *AhPDF1.1* increases the tolerance of *Arabidopsis halleri* to zinc excess (*Mirouze et al., 2006*). Overexpression of *AtPDF2.6* increases the resistance of *A. thaliana* to cadmium toxicity by reducing the content of cadmium ion in cytoplasm through chelation (*Luo et al., 2019a*). OsCAL1, a rice defensin-like proteins, is mainly expressed in parenchyma cells of root and xylem, and is involved in the cadmium transport and distribution in the aboveground, promoting the leaf cadmium accumulation (*Luo et al., 2018*). *CAL2* is the closest homolog of *CAL1*. *CAL2* is also cadmium chelating activity. Overexpression of *CAL2* increases the accumulation of cadmium in *Arabidopsis* and rice seedlings (*Luo et al., 2020*). Moreover, heterologous expression of *CAL2* enhanced cadmium sensitivity in

*Arabidopsis.* The heterologous overexpression of *BnaPDFL* plays a positive role in the cadmium tolerance of *Arabidopsis* (*Luo & Zhang, 2019*).

*Brassica napus* is a main oil crop species in the world. It can be used as human edible oil and animal protein feed, showing important economic value (*Raza, 2020*). *B. napus* ($A_n$ $A_n$ $C_n$ $C_n$, $2n = 4x = 38$) is a allotetraploid plant species formed by natural hybridization of two diploids, *B. rapa* ($A_r$ $A_r$, $2n = 2x = 20$) and *B. oleracea* ($C_o$ $C_o$, $2n = 2x = 18$ (*Bayer et al., 2017*)). Compared with *A. thaliana*, allopolyploid events in *B. napus* produce many repetitive fragments and homologous regions in the genome (*Pelé et al., 2017*).

At present, hundreds of PDFs have been isolated from *A. thaliana* (*Thomma, Cammue & Thevissen, 2002*), *Nicotiana alata* (*van der Weerden, Hancock & Anderson, 2010*), *Dahlia merkii* (*Thevissen et al., 2000*), and other plant species. However, genome-wide identification and comprehensive analysis of *PDF* family members has not been reported in *B. napus* yet. Therefore, bioinformatics methods were used to comprehensively analyze gene structure, phylogeny, and chromosomal distribution of genome-scale *PDFs*. In recent years, previous studies have shown that the PDFs participate in the adaptive response of plants to cadmium stress and salt stress (*Luo et al., 2018*; *Luo et al., 2019a*; *Khadka et al., 2020*; *Wu, Lin & Chuang, 2016*), but there are few studies on the response to nitrate limitation, ammonium excess, phosphorus starvation, and potassium deficiency. In addition, we also discussed gene expression and transcriptional responsive characteristics of *BnaPDFs* in different organs and diverse nutrient stresses, respectively. The genome-wide identification and characterization of *BnaPDFs* will enrich the understanding of their molecular characteristics, and will also provide elite gene resources for the genetic improvement of rapeseed resistance to nutrient stresses.

## MATERIALS AND METHODS

### Retrieval of gene sequences

In the Arabidopsis information resource (https://www.arabidopsis.org/), we obtained AtPDF amino acid sequences. Through a BLASTp analysis, the PDF protein sequences in *B. napus*, *B. rapa*, and *B. oleracea* were retrieved from the following database: The Arabidopsis Information Resource (TAIR10, https://www.arabidopsis.org/) for *A. thaliana*, the *Brassica* Database (BRAD) v. 1.1 (http://brassicadb.org/brad/) for *B. rapa* (*Wang et al., 2015*), Genoscope (http://www.genoscope.cns.fr/brassicanapus/) and *BnPIR* (http://cbi.hzau.edu.cn/bnapus/index.php) for *B. napus* (*Chalhoub et al., 2014*), *B. oleracea* v2.1 (http://plants.ensembl.org/ *Brassica oleracea*) for *B. oleracea* (*Yu et al., 2013*), National Center for Biotechnology Information (NCBI, https://www.ncbi.nlm.nih.gov/), Phytozome v.10 (http://phytozome.jgi.doe.gov/pz/portal.html) (*Goodstein et al., 2012*), and *EnsemblPlants* (http://plants.ensembl.org/index.html). All sequences with an E-value < $1e^{-10}$ were selected as candidate genes.

### Gene nomenclature

In this study, according to the nomenclature previously proposed, *BnaPDFs* were named according to the following criterion: genus name (one uppercase letter) + plant species

(two lowercase letters) + chromosome (followed by a period) + *PDF* homolog in *A. thaliana* (*Ostergaard & King, 2008*; *Li et al., 2015*). In this study, multiple homologs of *BnaPDFs* on the same chromosomes were named by serial numbering. For example, *BnaC2.PDF1.1a* and *BnaC2.PDF1.1b* indicated two At*PDF1.1* homologs on the chromosome C2 of *B. napus*.

## Physical mapping and family expansion analysis

Genomic annotations of *BnaPDFs* were used to determine their location and length on chromosomes (*Camacho et al., 2009*). We submitted *AtPDFs* and *BnaPDFs* at the start sites of chromosomes to MapGene2Chrom Web v2 (http://mg2c.iask.in/mg2c_v2.0/) online drawing tool to locate the physical location of AtPDFs and BnaPDFs. In this study, tandem repeats genes were defined as arrays of two or more *PDFs* within a genomic 100-kb region (*Smith & Waterman, 1987*).

## Sequence alignment and phylogeny analysis

To further analyze evolutionary relationships between *PDFs* of *B. napus* and *A. thaliana*, the PDF protein sequences of *B. napus* and *A. thaliana* were compared through performing multiple alignment of homologous protein sequences using ClustalW (*Thompson, Higgins & Gibson, 1994*). MEGA X (Molecular Evolutionary Genetics Analysis, http://www.megasoftware.net/) (*Kumar et al., 2018*) was used to construct a phylogenetic tree using the neighbor-joining method (*Saitou & Nei, 1987*). The parameters were set as follows: Bootstrap replications value of 1,000, *Poisson* correction, and complete deletion. Further, the online tool ITOL (https://itol.embl.de/) was used to edit and beautify the evolutionary tree.

## Analysis of evolutionary selection pressure and divergence of *BnaPDFs*

To understand selection pressure on *BnaPDFs* during the evolution process, synonymous (Ks) and non-synonymous (Ka) nucleotide substitution values and Ka/Ks of the duplicated gene pairs of *PDFs* were calculated (*Yang & Nielsen, 2000*). First, Clustal Omega (http://www.clustal.org/omega/) (*Sievers et al., 2011*) was used to pairwise alignment the amino acid sequences of AtPDFs and BnaPDFs. Subsequently, the Ka/Ks Calculator (*Wang et al., 2010*) (https://sourceforge.net/projects/kakscalculator2/) software was used to calculate the values of Ka, Ks, and Ka/Ks. According to Darwinian evolution, Ka/Ks > 1.0 is generally considered as positive selection, while Ka/Ks < 1.0 indicates purification selection occurs, and Ka/Ks = 1.0 means neutral selection. Ks was used to estimate the divergence time (T) according to the following formula: $T = Ks/2\lambda$. In this formula, $\lambda$ refers to the molecular replacement rate ($\lambda = 1.5 \times 10^{-8}$ for Brassicaceae species) (*Blanc & Wolfe, 2004*).

## Molecular characterization of BnaPDFs

ExPASy ProtoParam (https://web.expasy.org/protparam/) (*Wilkins et al., 1999*) was used to analyze physicochemical parameters of BnaPDFs, including amino acid number, molecular weight (MW, kD), theoretical isoelectric point (pI), grand average of

hydropathy (GRAVY), and instability index (II). A positive value of GRAVY indicates that the protein is hydrophobic, and a negative value is hydrophilic (*Kyte & Doolittle, 1982*). Instability index > 40.0 means the protein is unstable (*Guruprasad, Reddy & Pandit, 1990*).

The online website WoLF PSORT (http://www.genscript.com/wolf-psort.html) (*Horton et al., 2007*) was used to analyze subcellular localization of AtPDFs and BnaPDFs. The amino acid sequences of AtPDFs and BnaPDFs were submitted to the TMHMM v. 2.0 (http://www.cbs.dtu.dk/services/TMHMM/) program for transmembrane domain prediction. The online tool NetPhos 3.1 Server (http://www.cbs.dtu.dk/services/NetPhos/) was used to predict potential phosphorylation sites of AtPDFs and BnaPDFs (*Blom et al., 2004*).

The online SignalP v. 4.1 (http://www.cbs.dtu.dk/services/SignalP/) was used to analyze signal peptide sites of their amino acid positions in BnaPDFs (*Petersen et al., 2011*). To determine recombinant protein solubility of BnaPDFs, the recombinant protein solubility prediction (RPSP) (http://biotech.ou.edu) program was used to assume that recombinant proteins that were over-expressed in *Escherichia coli* (*Harrison & Bagajewicz, 2015*).

The STRING (Search Tool for Recurring Instances of Neighboring Genes) v 11.0 (https://string-db.org) program was used to retrieve PDF association networks (*Szklarczyk et al., 2019*). The online phyre2 (http://www.sbg.bio.ic.ac.uk/phyre2/webscripts/jobmonitor) program was used to predict three-dimensional structures of BnaPDFs (*Kelley et al., 2016*).

## Conserved motif identification of BnaPDFs

To further study conserved motifs of PDFs of *Arabidopsis* and *B. napus*, their protein sequences were submitted to the MEME (Multiple Expectation maximization for Motif Elicitation) v. 4.12.0 (http://meme-suite.org/tools/meme) program (*Bailey et al., 2009*). Motif length was set from 10–50 amino acid residues, and the maximum number of motifs was set at ten, with other parameters as default values. Finally, the online Weblogo (https://weblogo.berkeley.edu/logo.cgi) was used to display the conserved amino acid sequences in BnaPDFs (*Crooks et al., 2004*).

## Elucidation of gene structure and promoter regulatory *cis*-elements

The full-length genomic and CDS sequences of *PDF* family genes were obtained from *A. thaliana* and *B. napus* databases, and exon-intron structures of *PDFs* were displayed with Gene Structure Display Server (GSDS) (http://gsds.cbi.pku.edu.cn/). The 2.0 kb genomic sequence upstream of the start codon (ATG) of *BnaPDFs* and *AtPDFs* were downloaded in the *B. napus* Genome Browser (http://www.genoscope.cns.fr/brassicanapus/) and TAIR (https://www.arabidopsis.org/). These sequence files were submitted to PLACE v. 30.0 (http://www.dna.affrc.go.jp/PLACE/) program to predict promoter *cis*-acting regulatory DNA elements (*Higo et al., 1999*). The results were statistically classified, and the *cis*-acting elements of *PDFs* were visualized. In order to reveal the potential biological functions of *PDFs*, the R package GOseq was used for Gene Ontology (GO) analysis. In order to analyze miRNAs targeting *BnaPDF* genes, the

psRNATarget database (http://plantgrn.noble.org/psRNATarget/) was used to predict different miRNAs.

## Growth conditions

To further investigate transcriptional responses of *BnaPDFs* to various nutrient stresses, rapeseed seeds were selected for germination for 7 days, and the uniformly growing seedlings (cv. Zhongshuang 11) were transferred to a black plastic container with ten L of Hoagland nutrient solution. The basic nutrient solution contained 1.0 mM $KH_2PO_4$, 5.0 mM $KNO_3$, 5.0 mM $Ca(NO_3)_2 \cdot 4H_2O$, 2.0 mM $MgSO_4 \cdot 7H_2O$, 0.050 mM EDTA-Fe, 9.0 μM $MnCl_2 \cdot 4H_2O$, 0.80 μM $ZnSO_4 \cdot 7H_2O$, 0.30 μM $CuSO_4 \cdot 5H_2O$, 0.10 μM $Na_2MoO_4 \cdot 2H_2O$, and 46 μM $H_3BO_3$. To maintain ion concentrations in the nutrient solution, the solution was refreshed every 4 days. *B. napus* seedlings were cultured in the light chamber under the following growth conditions: light intensity of 200 μmol $m^{-2}$ $s^{-1}$, daytime temperature of 25 °C/night temperature of 22 °C, photoperiod 16 h light/8 h dark, and relative humidity of 70%.

Under treatment of nitrate ($NO_3^-$) deficiency, 7-day-old *B. napus* seedlings after seed germination were grown under high nitrate (6.0 mM $NO_3^-$) for 10 days, and then the plants were transferred to low nitrate (0.30 mM $NO_3^-$) for 3 d until sampling. Under treatment of ammonium ($NH_4^+$) toxicity, 7-day-old *B. napus* seedlings after seed germination were grown under high nitrate (6.0 mM $NO_3^-$) for 10 days, then were transferred to a nitrogen-free nutrient solution to grow for 3 d, and finally were grown for 6 h under excess ammonium (6.0 mM $NH_4^+$) until sampling. Under phosphate starvation treatment, 7-day-old uniformly growing *B. napus* seedlings after seed germination, which were grown under 250 μM phosphate ($KH_2PO_4$) for 10 days, were grown under five μM phosphate for 3 d until sampling. Under potassium deficiency treatment, 7-day-old uniformly growing *B. napus* seedlings after seed germination, which were first grown under high potassium (6.0 mM $K^+$) for 10 days, were then transferred to low potassium (0.03 mM $K^+$) for 3 d until sampling. Under salt stress treatment, 7-day-old uniformly growing *B. napus* seedlings after seed germination were cultured in NaCl-free nutrient solution for 10 days, and then were transferred to 200 mM NaCl for 1 d until sampling. Under cadmium (Cd) toxicity treatment, 7-day-old *B. napus* seedlings after seed germination are cultured in a Cd-free nutrient solution for 10 days, and then transferred to ten μM $CdCl_2$ for 12 h until sampling.

The shoots and roots of fresh rapeseed seedlings above-mentioned were sampled separately and were immediately stored at −80 °C. Each sample contained three independent biological replicates for transcriptional analysis of *BnaPDFs* under different nutrient stresses.

## Reverse transcription quantitative PCR assays

Total RNA of each sample was extracted by using pre-chilled TRIzol reagent (Invitrogen, Carlsbad, CA, USA) according to the manufacturer's recommendations. Purified total RNA was used as templates for cDNA synthesis using PrimeScript™ RT reagent Kit with gDNA Eraser (Perfect Real Time) (TaKaRa, Shiga, Japan), and then quantitative PCR

**Table 1 Copy number of the *Plant Defensins* (*PDF*) genes in four *Brassica* species.**

| Item | Brassica napus (1,130 Mb) | Brassica rapa (465 Mb) | Brassica oleracea (485 Mb) | Arabidopsis thaliana (125 Mb) |
|---|---|---|---|---|
| *PDF1* | 25 | 10 | 6 | 7 |
| *PDF2* | 12 | 4 | 5 | 6 |
| Total | 37 | 14 | 11 | 13 |

assays were used to analyze the relative expression of *BnaPDFs*. The PCR program was set as follows: 95 °C for 3 min, 95°C for 10 s, 40 cycles, and 60 °C for 30 s (*Hua et al., 2018*). *BnaEF1-α* and *BnaGDI1* were used as internal references, and expression levels of *BnaPDFs* were calculated using the $2^{-\Delta\Delta C_T}$ method (*Livak & Schmittgen, 2001*). Each sample contained three independent biological replicates, and each cDNA template had three technical replicates.

### Statistical data analysis

All data were presented as mean ± *SD*. Comparisons among different treatments were performed using Student's *t*-test or one-way ANOVA. A value of $p < 0.05$ was considered statistically significant. Pearson correlation and statistical analysis were carried out using GraphPad Prism 8.0 software.

## RESULTS

### Genome-wide identification of *PDFs* in Brassica species

In *A. thaliana*, the *PDF* family included three subfamilies consisting of 13 members, including *AtPDF1s* (*AtPDF1.1*, *AtPDF1.2a*, *AtPDF1.2b*, *AtPDF1.2c*, *AtPDF1.3*, *AtPDF1.4*, and *AtPDF1.5*), and *AtPDF2s* (*AtPDF2.1*, *AtPDF2.2*, *AtPDF2.3*, *AtPDF2.4*, *AtPDF2.5*, and *AtPDF2.6*). Subsequently, we identified a total of 17, 16, and 44 *PDF* homologs in *B. rapa*, *B. oleracea*, and *B. napus*, respectively (Table 1). In detail, the number of *BnaPDF* family members ranged from one (*BnaPDF1.2c* and *BnaPDF1.3*) to nine (*BnaPDF1.2b*). However, no homologous sequences of *AtPDF1.5*, *AtPDF2.1*, and *AtPDF2.4* were identified in *B. napus*. It suggested that they might have been lost in the evolution of *Brassica* species due to functional redundancy. Also, they could loss due to genetic drift or loss during duplication process.

### Genomic distribution and expansion patterns of *BnaPDFs*

Chromosomal locations of *PDFs* on their respective chromosomes were listed in Fig. 1. Chromosomal location analysis of *BnaPDFs* revealed that the *PDF* family were distributed in the *B. napus* genome. A total of 37 *BnaPDFs* were distributed on 13 chromosomes, and 17 *BnaPDFs* were identified to be located on six chromosomes (A2, A5, A6, A7, A8, and A9) of the A subgenome. Likewise, 20 *BnaPDFs* are identified on seven chromosomes (C1, C2, C4, C6, C7, C8, and C9) of the C subgenome (Table 2, Fig. 1B).

Gene amplification is used as a main driving force for adaptive evolution of species (*Tang et al., 2010*; *Jiao et al., 2011*). Gene amplification mainly includes tandem

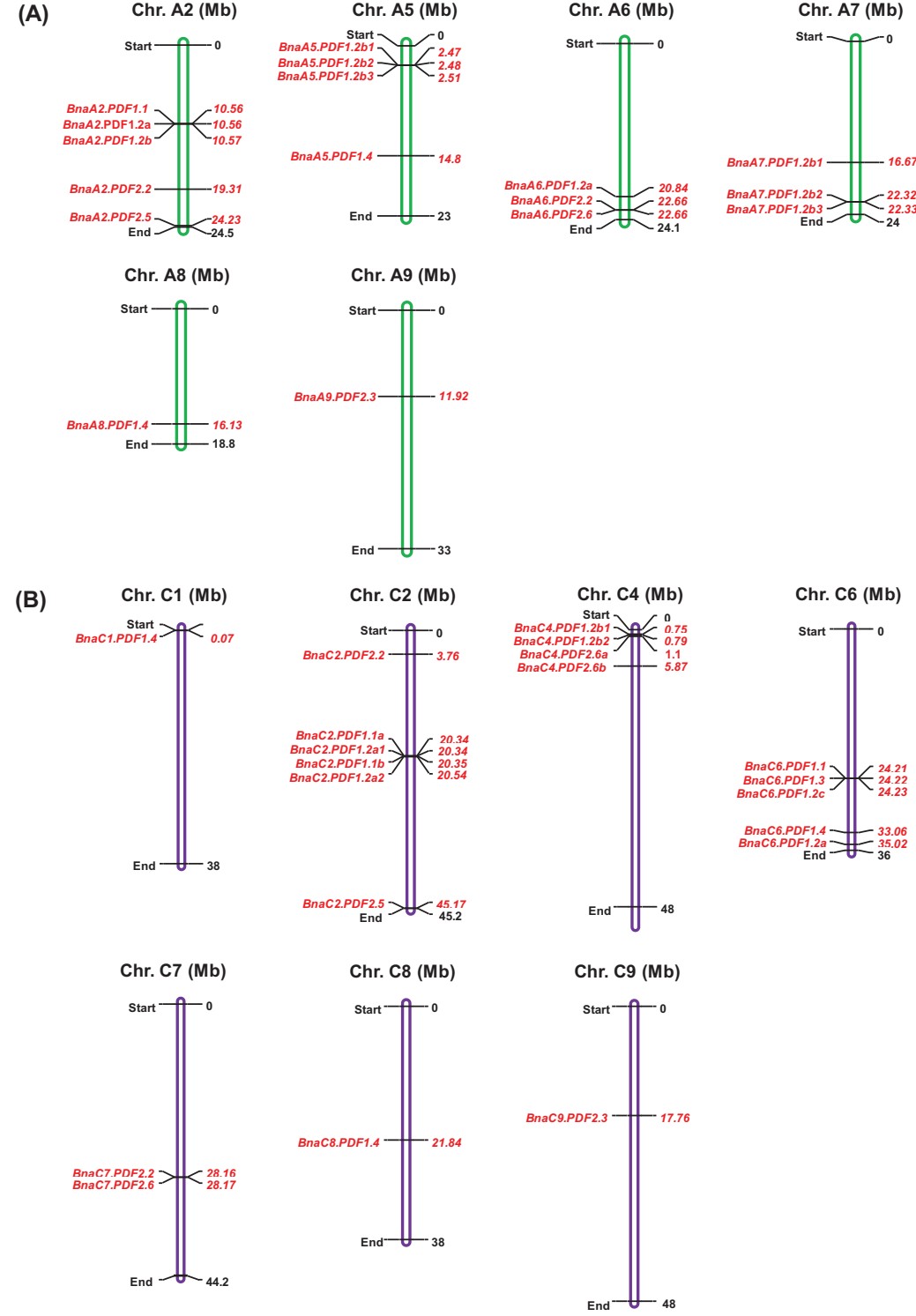

**Figure 1 Physical mapping of the _plant defensin_ (_PDF_) family members in _Brassica napus_.** The _BnaPDFs_ were physically mapped onto 13 chromosomes of _B. napus_, (A) A subgenome: A2, A5, A6, A7, A8 and A9; (B) C subgenome: C1, C2, C4, C6, C7, C8, and C9.

**Table 2 Molecular characterization of the *Plant Defensins* (*PDF*) genes in *Arabidopsis thaliana* and *Brassica napus*.**

| Gene ID | Gene name | Block | CDS (bp) | Exon/ intron | Amino acid (aa) | Ka | Ks | Ka/Ks | Divergent time (Mya) |
|---|---|---|---|---|---|---|---|---|---|
| BnaA02g17500D | *BnaA2.PDF1.1* | E | 243 | 2/1 | 80 | 0.1772 | 0.4305 | 0.4116 | 14.35 |
| BnaC02g23400D | *BnaC2.PDF1.1a* | E | 243 | 2/1 | 80 | 0.0586 | 0.3035 | 0.1930 | 10.11 |
| BnaC02g23420D | *BnaC2.PDF1.1b* | E | 243 | 2/1 | 80 | 0.1775 | 0.4289 | 0.4137 | 14.29 |
| BnaC06g22110D | *BnaC6.PDF1.1* | E | 243 | 2/1 | 80 | 0.0993 | 0.3175 | 0.3128 | 10.58 |
| BnaA02g17510D | *BnaA2.PDF1.2a* | V | 249 | 2/1 | 82 | 0.2324 | 0.5643 | 0.4118 | 18.81 |
| BnaA06g30850D | *BnaA6.PDF1.2a* | V | 243 | 2/1 | 80 | 0.2157 | 0.4763 | 0.4529 | 15.87 |
| BnaC02g23410D | *BnaC2.PDF1.2a1* | V | 249 | 2/1 | 82 | 0.2313 | 0.4939 | 0.4684 | 16.46 |
| BnaC02g23620D | *BnaC2.PDF1.2a2* | V | 258 | 2/1 | 85 | 0.4893 | 0.7062 | 0.6929 | 23.54 |
| BnaC06g36530D | *BnaC6.PDF1.2a* | V | 243 | 2/1 | 80 | 0.0947 | 0.4695 | 0.2018 | 15.65 |
| BnaA02g17520D | *BnaA2.PDF1.2b* | I | 243 | 2/1 | 80 | 0.0749 | 0.3831 | 0.1956 | 12.77 |
| BnaA05g04680D | *BnaA5.PDF1.2b1* | I | 243 | 2/1 | 80 | 0.5090 | 0.5004 | 1.0172 | 16.68 |
| BnaA05g04690D | *BnaA5.PDF1.2b2* | I | 246 | 2/1 | 81 | 0.4842 | 0.4739 | 1.0218 | 15.79 |
| BnaA05g04760D | *BnaA5.PDF1.2b3* | I | 243 | 2/1 | 80 | 0.4710 | 0.5987 | 0.7866 | 19.95 |
| BnaA07g21570D | *BnaA7.PDF1.2b1* | I | 243 | 2/1 | 80 | 0.1011 | 0.3673 | 0.2753 | 12.24 |
| BnaA07g32130D | *BnaA7.PDF1.2b2* | I | 243 | 2/1 | 80 | 0.0853 | 0.3282 | 0.2600 | 10.94 |
| BnaA07g32150D | *BnaA7.PDF1.2b3* | I | 240 | 2/1 | 79 | 0.0705 | 0.3482 | 0.2023 | 11.60 |
| BnaC04g53180D | *BnaC4.PDF1.2b1* | I | 243 | 2/1 | 80 | 0.5064 | 0.5078 | 0.9972 | 16.92 |
| BnaC04g53190D | *BnaC4.PDF1.2b2* | I | 243 | 2/1 | 80 | 0.4662 | 0.6050 | 0.7706 | 20.16 |
| BnaC06g22140D | *BnaC6.PDF1.2c* | V | 240 | 2/1 | 79 | 0.0952 | 0.2719 | 0.3502 | 9.06 |
| BnaC06g22120D | *BnaC6.PDF1.3* | I | 240 | 2/1 | 79 | 0.0868 | 0.3933 | 0.2206 | 13.11 |
| BnaA05g19430D | *BnaA5.PDF1.4* | B | 288 | 3/2 | 95 | 0.1206 | 0.5015 | 0.2405 | 16.72 |
| BnaA08g22000D | *BnaA8.PDF1.4* | B | 237 | 2/1 | 78 | 0.4702 | 3.1360 | 0.1499 | 104.53 |
| BnaC01g00250D | *BnaC1.PDF1.4* | B | 231 | 2/1 | 76 | 0.4451 | 3.0059 | 0.1481 | 100.19 |
| BnaC06g33060D | *BnaC6.PDF1.4* | B | 267 | 2/1 | 88 | 0.3904 | 5.6153 | 0.0695 | 187.17 |
| BnaC08g18880D | *BnaC8.PDF1.4* | B | 237 | 2/1 | 78 | 0.0465 | 0.2858 | 0.1626 | 9.52 |
| BnaA02g26210D | *BnaA2.PDF2.2* | K | 234 | 2/1 | 77 | 0.0572 | 0.5222 | 0.1096 | 17.40 |
| BnaA06g34320D | *BnaA6.PDF2.2* | K | 234 | 2/1 | 77 | 0.0505 | 0.2948 | 0.1713 | 9.82 |
| BnaC02g47680D | *BnaC2.PDF2.2* | K | 234 | 2/1 | 77 | 0.0572 | 0.5903 | 0.0969 | 19.67 |
| BnaC07g21510D | *BnaC7.PDF2.2* | K | 231 | 2/1 | 76 | 0.0525 | 0.3780 | 0.1390 | 12.60 |
| BnaA09g18990D | *BnaA9.PDF2.3* | K | 234 | 2/1 | 77 | 0.0743 | 0.3135 | 0.2371 | 10.45 |
| BnaC09g20670D | *BnaC9.PDF2.3* | K | 234 | 2/1 | 77 | 0.0826 | 0.3291 | 0.2511 | 10.97 |
| BnaA02g33850D | *BnaA2.PDF2.5* | X | 222 | 2/1 | 73 | 0.1430 | 0.8075 | 0.1771 | 26.91 |
| BnaC02g42620D | *BnaC2.PDF2.5* | X | 222 | 2/1 | 73 | 0.1452 | 0.7109 | 0.2043 | 23.69 |
| BnaA06g34310D | *BnaA6.PDF2.6* | K | 222 | 2/1 | 73 | 0.1761 | 0.3469 | 0.5076 | 11.56 |
| BnaC04g53620D | *BnaC4.PDF2.6a* | K | 222 | 2/1 | 73 | 0.4993 | 1.5605 | 0.3200 | 52.01 |
| BnaC04g07810D | *BnaC4.PDF2.6b* | K | 222 | 2/1 | 73 | 0.4988 | 1.7036 | 0.2928 | 56.78 |
| BnaC07g21520D | *BnaC7.PDF2.6* | K | 222 | 2/1 | 73 | 0.1744 | 0.3814 | 0.4573 | 12.71 |
| AT1G75830 | *AtPDF1.1* | E | 243 | 2/1 | 80 | | | | |
| AT1G75830 | *AtPDF1.2a* | V | 243 | 2/1 | 80 | | | | |
| AT2G26020 | *AtPDF1.2b* | I | 243 | 2/1 | 80 | | | | |
| AT5G44430 | *AtPDF1.2c* | V | 243 | 2/1 | 80 | | | | |

(Continued)
| Table 2 (continued) | | | | | | | | | |
|---|---|---|---|---|---|---|---|---|---|
| Gene ID | Gene name | Block | CDS (bp) | Exon/ intron | Amino acid (aa) | Ka | Ks | Ka/Ks | Divergent time (Mya) |
| AT2G26010 | *AtPDF1.3* | I | 243 | 2/1 | 80 | | | | |
| AT1G19610 | *AtPDF1.4* | B | 237 | 2/1 | 78 | | | | |
| AT1G55010 | *AtPDF1.5* | C | 243 | 2/1 | 80 | | | | |
| AT2G02120 | *AtPDF2.1* | K | 234 | 2/1 | 77 | | | | |
| AT2G02100 | *AtPDF2.2* | K | 234 | 2/1 | 77 | | | | |
| AT2G02130 | *AtPDF2.3* | K | 234 | 2/1 | 77 | | | | |
| AT1G61070 | *AtPDF2.4* | D | 231 | 2/1 | 76 | | | | |
| AT5G63660 | *AtPDF2.5* | X | 222 | 2/1 | 73 | | | | |
| AT2G02140 | *AtPDF2.6* | K | 222 | 2/1 | 73 | | | | |

**Note:**
CDS, coding sequence; Ka, non-synonymous nucleotide substitution rate; Ks, synonymous nucleotide substitution rate.

duplication, segmental duplication, whole-genome duplication/polyploidization, and replicative transposition (*Freeling, 2009*). A total of twenty-four of *BnaPDFs* existed in the form of tandem duplication gene clusters, which indicated that tandem duplication might be a primary way of the *BnaPDF* family amplification. The Arabidopsis genome can be divided into 24 ancestral cruciferous blocks, labeled A–X (*Schranz, Lysak & Mitchell-Olds, 2006*). Table 2 showed that *AtPDFs* and their corresponding *BnaPDF* homologs were located on the same chromosomal blocks. In detail, the PDFs were located on ten chromosomal blocks, including B, C, D, E, I, K, S, U, V, X.

## Phylogeny analysis of BnaPDFs

To reveal molecular and phylogenetic relationships between the PDFs of *B. napus* and *A. thaliana*, protein sequences from BnaPDFs and AtPDFs were used to construct unrooted phylogenetic trees (Fig. 2). In Arabidopsis, PDF family members are mainly divided into two evolutionary branches: PDF1s and PDF2s. Besides, a phylogenetic analysis of 13 PDFs of *A. thaliana* and 37 PDFs of *B. napus* was performed. As shown in Fig. 2, the phylogenetic tree was clearly clustered into two well-supported classes, namely PDF1 and PDF2. In each evolutionary clade, BnaPDF members were clustered closely together with the corresponding homolog in Arabidopsis. The results showed that PDFs had diverged before the formation of *Brassica* species. Most of the PDFs in each subfamily had short branch lengths (Fig. 2), suggesting that genetic differentiation has occurred recently.

## Molecular characterization of BnaPDFs

Physicochemical properties of PDF family proteins were analyzed by the ExPASy online software. The results showed that CDS lengths of *BnaPDFs* stretched from 222 bp (BnaPDF2.5s and BnaPDF2.6s) to 288 bp (BnaA5.PDF1.4), and protein lengths of PDFs were also identified to be varied between 73 and 95 amino acids. In compliance with protein lengths, molecular weights of PDFs also differed ranging from 7.6 to 10.5 kD (Table 2). Isoelectric points (pI) of PDFs ranged from 5.01 (*BnaC1.PDF1.4*) and 9.82

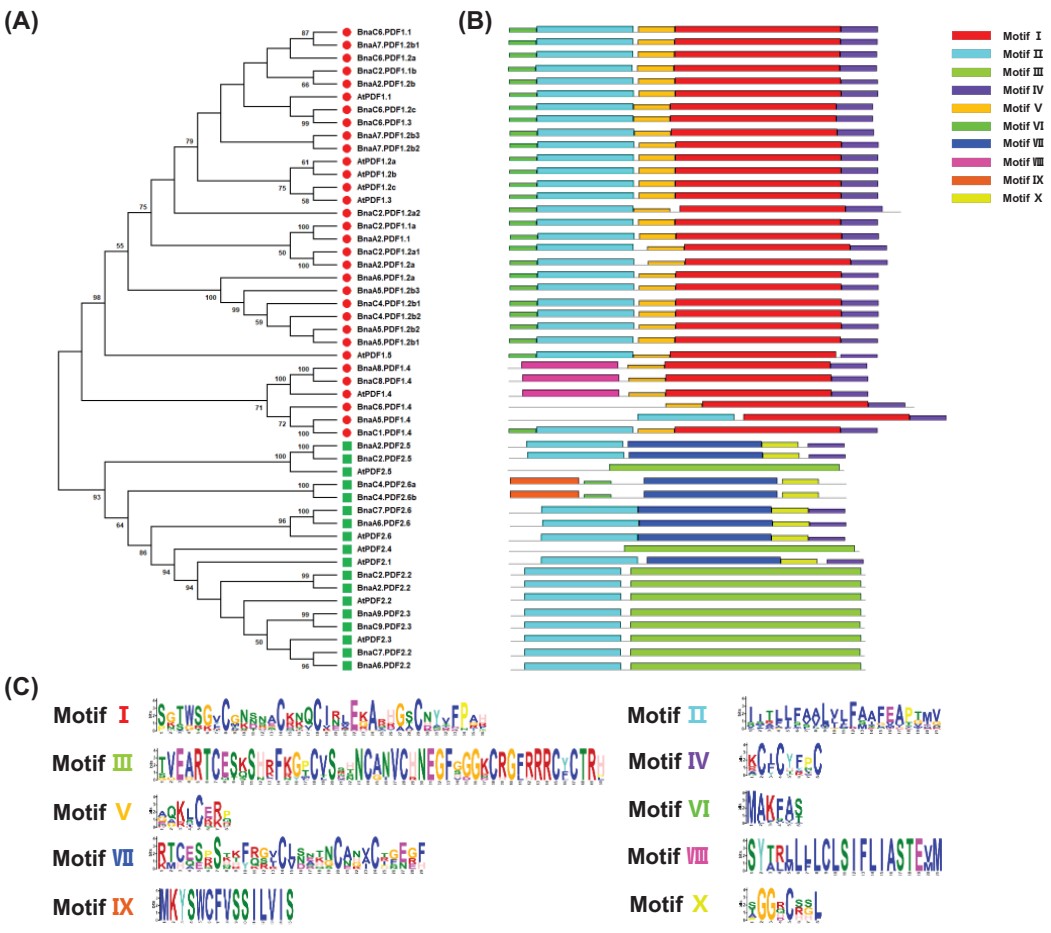

**Figure 2 Phylogeny analysis and conserved motifs of the plant defensins (PDFs) in *Arabidopsis thaliana* and *Brassica napus*.** (A) Phylogeny analysis of AtPDFs and BnaPDFs. The PDF protein sequence was multiple-aligned using ClustalW, and then the neighboring method in the MEGA X was used to construct a phylogenetic tree. In the phylogenetic tree, related taxa are clustered together (1,000 repeats), The evolution distance is calculated using the poisson correction method, and the unit is the number of amino acid substitutions at each position. Molecular identification (B) and sequence characterization (C) of the conserved motifs in the PDFs. In (B) the 10 different colored boxes represent ten conserved motifs (patterns I–X), and grey lines represent without conserved regions. In (C) the larger the fonts, the more conserved the motifs.

(BnaA9.PDF2.3 and BnaC9.PDF2.3). The pIs of most PDF proteins were above seven, except for BnaC1.PDF1.4, BnaC4.PDF2.6a, BnaC4.PDF2.6b (Table 3). In terms of instability index (IIs), 51.35% (19/37) of BnaPDFs with IIs greater than 40 was presumed to be stable proteins, and the other part is weakly stable proteins. Grand average of hydropathy index (GRAVY) of BnaA6.PDF1.2a, BnaA5.PDF1.2b2, BnaPDF1.4, BnaC7. PDF2.2, and BnaPDF2.5 was negative (Table 3), the results showed that the PDF family proteins are hydrophilic proteins.

The Wolf Psort software was used to predict subcellular locations of BnaPDFs and AtPDFs (Table 3). The prediction results indicated that PDF1s and PDF2s of Arabidopsis were potentially located in the extracellular matrix. The subcellular localization of BnaPDFs was predicted to have the same position.

**Table 3 Molecular characterization of the *Plant Defensins* (*PDF*) in *Arabidopsis thaliana* and *Brassica napus*.**

| Gene ID | Gene name | pI | MW (kDa) | II | Aliphatic inedx | GRAVY | TM domains | Subcellular localization |
|---------|-----------|-----|----------|-----|-----------------|-------|------------|--------------------------|
| BnaA02g17500D | *BnaA2.PDF1.1* | 9.00 | 8918.54 | 32.13 | 71.00 | 0.065 | 1 | Extr |
| BnaC06g22110D | *BnaC6.PDF1.1* | 8.70 | 8764.31 | 30.55 | 74.50 | 0.259 | 1 | Extr |
| BnaC02g23400D | *BnaC2.PDF1.1a* | 8.47 | 8700.26 | 30.55 | 83.12 | 0.348 | 1 | Extr |
| BnaC02g23420D | *BnaC2.PDF1.1b* | 9.01 | 8946.55 | 33.19 | 71.00 | 0.057 | 1 | Extr |
| BnaA02g17510D | *BnaA2.PDF1.2a* | 9.24 | 9240.08 | 43.41 | 73.90 | 0.237 | 1 | Extr |
| BnaA06g30850D | *BnaA6.PDF1.2a* | 9.41 | 9232.09 | 45.29 | 65.88 | −0.005 | 1 | Extr |
| BnaC02g23410D | *BnaC2.PDF1.2a1* | 9.24 | 9240.08 | 43.41 | 73.90 | 0.237 | 1 | Extr |
| BnaC02g23620D | *BnaC2.PDF1.2a2* | 8.47 | 9145.64 | 38.52 | 73.65 | 0.148 | 1 | Extr |
| BnaC06g36530D | *BnaC6.PDF1.2a* | 8.70 | 8778.33 | 29.49 | 75.75 | 0.295 | 1 | Extr |
| BnaA02g17520D | *BnaA2.PDF1.2b* | 8.47 | 8718.23 | 32.96 | 73.38 | 267 | 1 | Extr |
| BnaA05g04680D | *BnaA5.PDF1.2b1* | 9.18 | 9223.98 | 52.62 | 70.75 | 0.075 | 1 | Extr |
| BnaA05g04690D | *BnaA5.PDF1.2b2* | 9.20 | 9382.06 | 72.77 | 61.48 | −0.101 | 1 | Extr |
| BnaA05g04760D | *BnaA5.PDF1.2b3* | 9.06 | 9185.82 | 73.26 | 62.25 | 0.025 | 1 | Extr |
| BnaA07g21570D | *BnaA7.PDF1.2b1* | 8.70 | 8820.41 | 29.49 | 78.12 | 0.294 | 1 | Extr |
| BnaA07g32130D | *BnaA7.PDF1.2b2* | 8.70 | 8830.37 | 29.45 | 78.12 | 0.336 | 1 | Extr |
| BnaA07g32150D | *BnaA7.PDF1.2b3* | 8.15 | 8555.01 | 30.42 | 76.71 | 0.411 | 1 | Extr |
| BnaC04g53180D | *BnaC4.PDF1.2b1* | 9.03 | 9164.87 | 67.38 | 70.75 | 0.033 | 1 | Extr |
| BnaC04g53190D | *BnaC4.PDF1.2b2* | 9.18 | 9251.99 | 63.15 | 68.25 | 0.04 | 1 | Extr |
| BnaC06g22140D | *BnaC6.PDF1.2c* | 8.49 | 8599.07 | 28.57 | 66.84 | 0.363 | 1 | Extr |
| BnaC06g22120D | *BnaC6.PDF1.3* | 8.47 | 8659.12 | 33.36 | 63.16 | 0.209 | 1 | Extr |
| BnaA05g19430D | *BnaA5.PDF1.4* | 8.67 | 10567.17 | 52.94 | 67.79 | −0.028 | 1 | Extr |
| BnaA08g22000D | *BnaA8.PDF1.4* | 8.85 | 8814.28 | 57.46 | 60.13 | 0.103 | 1 | Extr |
| BnaC01g00250D | *BnaC1.PDF1.4* | 5.01 | 8335.5 | 30.87 | 79.61 | 0.312 | 0 | Extr |
| BnaC06g33060D | *BnaC6.PDF1.4* | 8.82 | 10136.95 | 45.86 | 73.30 | −0.13 | 1 | Extr |
| BnaC08g18880D | *BnaC8.PDF1.4* | 8.85 | 8844.3 | 53.33 | 58.85 | 0.071 | 1 | Extr |
| BnaA02g26210D | *BnaA2.PDF2.2* | 9.33 | 8511.05 | 49.43 | 68.31 | 0.048 | 1 | Extr |
| BnaA06g34320D | *BnaA6.PDF2.2* | 9.63 | 8510.06 | 34.44 | 61.95 | 0.032 | 1 | Extr |
| BnaC02g47680D | *BnaC2.PDF2.2* | 9.33 | 8497.02 | 46.93 | 68.31 | 0.048 | 1 | Extr |
| BnaC07g21510D | *BnaC7.PDF2.2* | 9.49 | 8382.83 | 35.88 | 57.63 | −0.012 | 1 | Extr |
| BnaA09g18990D | *BnaA9.PDF2.3* | 9.82 | 8691.31 | 49.86 | 63.25 | −0.069 | 1 | Extr |
| BnaC09g20670D | *BnaC9.PDF2.3* | 9.82 | 8707.35 | 50.84 | 68.31 | 0.001 | 1 | Extr |
| BnaA02g33850D | *BnaA2.PDF2.5* | 9.14 | 8414.81 | 61.47 | 69.45 | −0.036 | 1 | Extr |
| BnaC02g42620D | *BnaC2.PDF2.5* | 9.14 | 8414.81 | 61.47 | 69.45 | −0.036 | 1 | Extr |
| BnaA06g34310D | *BnaA6.PDF2.6* | 8.71 | 7687.19 | 37.47 | 86.85 | 0.495 | 1 | Extr |
| BnaC04g53620D | *BnaC4.PDF2.6a* | 6.49 | 8605.98 | 37.43 | 64.11 | 0.034 | 1 | Extr |
| BnaC04g07810D | *BnaC4.PDF2.6b* | 6.49 | 8605.98 | 37.43 | 64.11 | 0.034 | 1 | Extr |
| BnaC07g21520D | *BnaC7.PDF2.6* | 8.71 | 7687.19 | 37.47 | 86.85 | 0.495 | 1 | Extr |
| AT1G75830 | *AtPDF1.1* | 8.47 | 8709.22 | 27.49 | 74.5 | 0.339 | 1 | Extr |
| AT1G75830 | *AtPDF1.2a* | 8.14 | 8518.03 | 27.68 | 81.88 | 0.454 | 1 | Extr |
| AT2G26020 | *AtPDF1.2b* | 8.14 | 8640.16 | 28.23 | 78.12 | 0.389 | 1 | Extr |
| AT5G44430 | *AtPDF1.2c* | 8.14 | 8550.03 | 25.27 | 75.75 | 0.358 | 1 | Extr |

| Gene ID | Gene name | pI | MW (kDa) | II | Aliphatic inedx | GRAVY | TM domains | Subcellular localization |
|---|---|---|---|---|---|---|---|---|
| AT2G26010 | *AtPDF1.3* | 8.14 | 8580.1 | 25.27 | 78.25 | 0.436 | 1 | Extr |
| AT1G19610 | *AtPDF1.4* | 8.44 | 8840.28 | 55.88 | 52.69 | 0.113 | 1 | Extr |
| AT1G55010 | *AtPDF1.5* | 5.65 | 9139.47 | 31.07 | 65.88 | 0.006 | 0 | Extr |
| AT2G02120 | *AtPDF2.1* | 9.14 | 8578.05 | 41.13 | 56.88 | 0.017 | 1 | Extr |
| AT2G02100 | *AtPDF2.2* | 9.37 | 8524.07 | 40.13 | 63.25 | 0.201 | 1 | Extr |
| AT2G02130 | *AtPDF2.3* | 9.63 | 8544.07 | 39.18 | 59.48 | 0.004 | 1 | Extr |
| AT1G61070 | *AtPDF2.4* | 8.52 | 8289.68 | 51.47 | 80.79 | 0.182 | 1 | Extr |
| AT5G63660 | *AtPDF2.5* | 8.92 | 8387.69 | 56.73 | 54.79 | −0.13 | 0 | Extr |
| AT2G02140 | *AtPDF2.6* | 8.92 | 7718.14 | 76.05 | 73.56 | 0.281 | 1 | Extr |

**Note:**
GRAVY, grand average of hydropathy; II, instability index; MW, molecular weight; pI, isoelectric point; Extr, extracellular region.

Phosphorylation, a ubiquitous protein activity regulation mechanism in organisms, plays an important role in cell signal transduction and usually occurs at the sites of serine, threonine, and tyrosine (*Mijakovic, Grangeasse & Turgay, 2016*). The NetPhos 3.1 software was used to analyze the phosphorylation sites of AtPDF and BnaPDF proteins (Fig. S1). The PDF proteins had differential preference for phosphorylation on serine, threonine, and tyrosine sites. The phosphorylation sites of most PDFs were mainly serine and threonine residues.

Transmembrane domains are main sites where proteins in the membrane combine with lipid (*Kim et al., 2020*). AtPDF1.5, AtPDF2.5, and BnaC1.PDF1.4 were predicted to have no transmembrane domains, and the remaining BnaPDF and AtPDF proteins contained a transmembrane domain. Therefore, most PDF family members functioned on the cell membrane (Fig. S2).

The members of BnaPDFs and AtPDFs contained a signal peptides, which guided the newly synthesized protein to secret the organelles and performed functions.

Recombinant protein solubility prediction showed that the recombinant BnaC2. PDF1.1b, BnaC2.PDF1.2a1, BnaA2.PDF1.2a, BnaA2.PDF1.2b, and BnaA5.PDF1.4 expressed in plant might be unstable, whereas the rest of BnaPDF recombinant proteins expressed in plant showed strong stability (Table 3).

### Identification of evolutionary selection pressure on *BnaPDFs*

Selection pressure analysis also reflect differentiation of gene families in the process of evolution. Ratios of substitution (Ka) and synonymous substitution (Ks) were used to estimate selection pressure on *BnaPDFs* during evolution (Table 2). The Ka values of *BnaPDFs* ranged from 0.0465 (*BnaC8.PDF1.4*) to 0.5090 (*BnaA5.PDF1.2b1*) with an average value of 0.2312. The Ks values of *BnaPDFs* ranged from 0.1648 (*BnaA8.PDF1.4*) to 5.6153 (*BnaC6.PDF1. 4*) with average value of 0.8204. The Ka/Ks ratios of 34 *BnaPDFs* were between 0.0695 and 0.7866, which indicated that *BnaPDFs* had undergone strong negative evolution to preserve their functionality. The ratio of Ka/Ks of *BnaC4.PDF1.2b1*

was 0.9972, which indicated *BnaC4.PDF1.2b1* underwent neutral selection. The ratios of Ka/Ks of *BnaA5.PDF1.2b1* and *BnaA5.PDF1.2b2* were greater than one, indicating that these two genes were subject to positive selection during evolutionary selection.

The separation of Arabidopsis and *Brassica* species occurred 12 to 20 million years ago (Mya). The Ks value results showed that duplication events of most *PDFs* occurred 10–100 Mya, which indicated that the species formation of *Brassica* plants might be accompanied by the functional divergence of *PDF* genes (Fig. 2A).

## Conserved domain, gene structure, protein interaction and transcriptional regulatory analysis

The MEME program was used to analyze the conservation of all PDF proteins, and ten conservative motifs were found (Fig. 2B). This study showed that each PDF contained one to five motifs. Motif II were highly conserved in all PDF sequences, except for several PDF family members (BnaA8.PDF1.4, BnaC8.PDF1.4, BnaC6.PDF1.4, BnaC4.PDF2.6a, BnaC4.PDF2.6b, AtPDF1.4, AtPDF2.4 and AtPDF2.5) (Fig. 2B). The composition patterns of protein conserved motifs in each subgroup were similar, indicating that the same subgroup might had similar biological functions. Each subfamily had its specific conserved motif. All PDF1 subfamily members include motif I, motif IV and motif V. Motif I, Motif V, and Motif VI were unique to the PDF1 subfamily, and Motif III and Motif VII were unique to the PDF2 subfamily (Fig. 2B).

The exon-intron structure of *BnaPDFs* was determined by comparing the genomic sequences and its corresponding coding sequences (Fig. 3). The statistical results showed that except *AtPDF1.3* had only one exon, *BnaA5.PDF1.4* contained three exons and two introns, and all other genes contained only two exons and one intron. In addition, several *BnaPDF* genes contained longer intron structures, especially for, *BnaC4.PDF1.2b1*, *BnaA6.PDF2.6*.

Phyre2 was used to predict secondary and three-dimensional structures of BnaPDFs (Fig. S4). Secondary structures of most BnaPDFs are mainly composed of alpha helix, beta strand, disordered and transmembrane helix. Alpha helix was the main component of BnaPDF secondary structures, accounting for 30% (BnaC4.PDF2.6) to 60% (BnaC6.PDF1.4), with an average value of 52%. The proportion of disordered structures ranged from 25% (BnaC4.PDF1.2b1) to 46% (BnaA8.PDF1.4), with an average of 35%, whereas the proportion of β-turns and transmembrane helix was relatively small. The three-dimensional structures of BnaPDFs, formed by further winding and folding on the basis of secondary structure, were mainly composed of alpha helix and disordered structures.

PDF proteins interaction analysis results showed that *PDFs* showed closely interaction with each other (Fig. 4). Almost all PDF proteins were associated with innate immune proteins (such as disease-related proteins) and the DEFL-encoding proteins. From PDF interaction networks involving PDF1.3, PDF2.1, PDF2.3, and PDF2.5, the genes regulating plant stress response, phosphorylation-mediated growth and development, ubiquitination-responsive gene expression, and hormone-mediated signals were highly enriched (Fig. 4, Fig. S5).

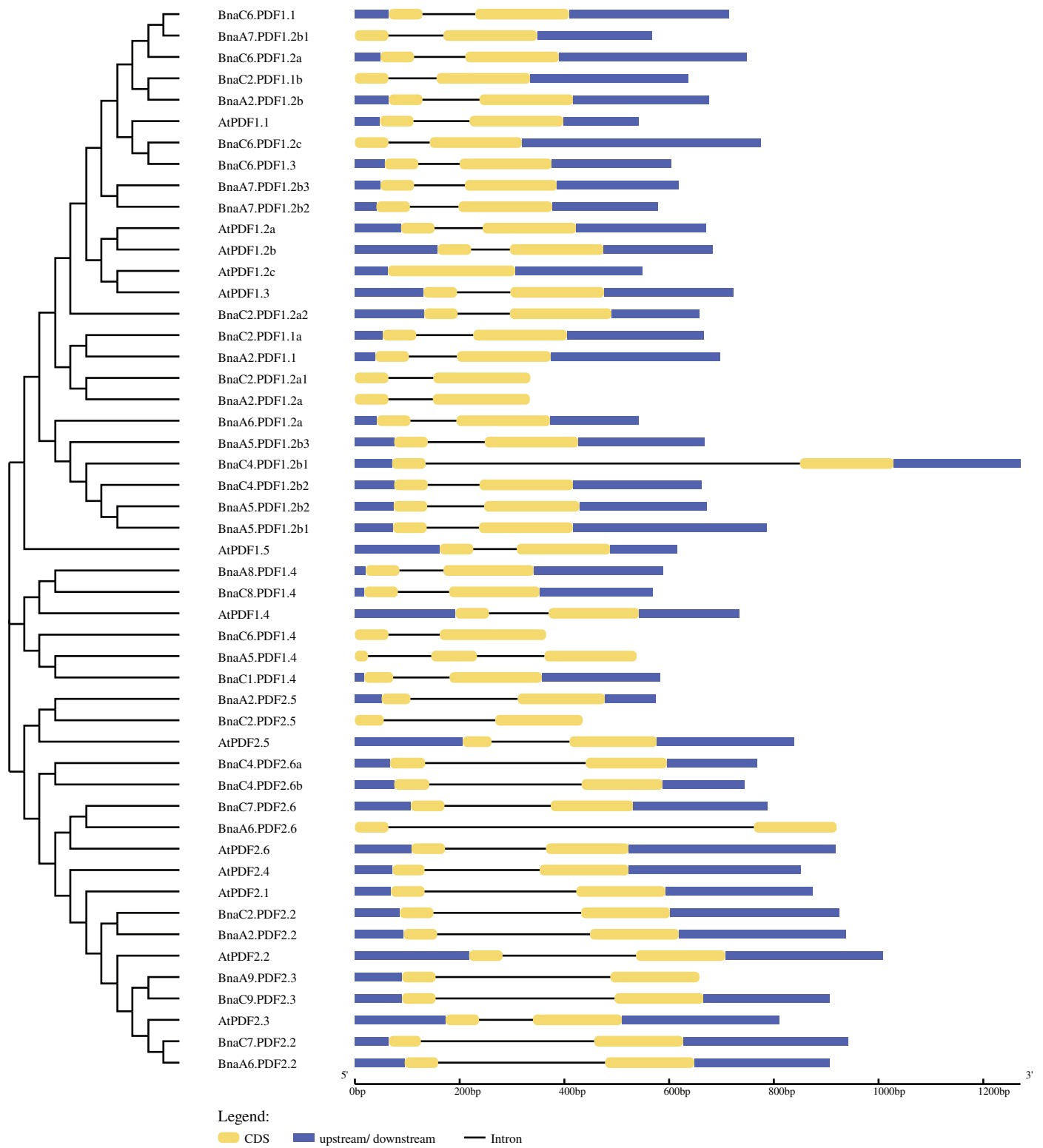

**Figure 3  Gene structures of the *PDF* family genes in *Arabidopsis thaliana* and *Brassica napus*.** In the GSDS server, the exon-intron structures of the *PDFs* were determined by comparing the coding sequence with the corresponding genome sequence. The yellow boxes represent exons, blue boxes indicate upstream or downstream, and the lines represent introns. The diagram was obtained using the GSDS web server.

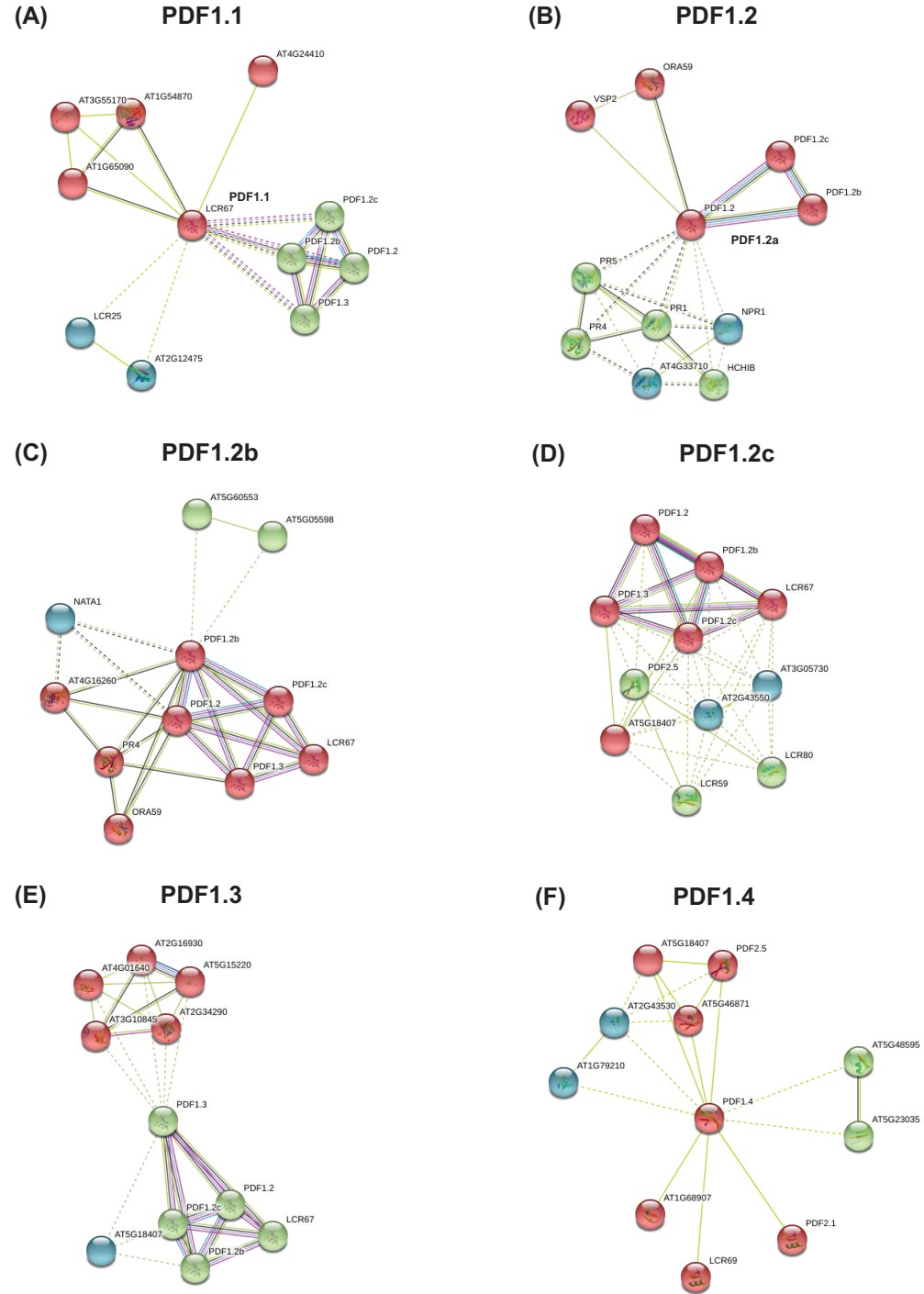

**Figure 4 The protein-protein interaction network of plant defensins.** Constructed PDF1.1 (A), PDF1.2a (B), PDF1.2b (C), PDF1.2c (D), PDF1.3 (E), PDF1.4 (F) and other protein interaction networks provided by the STRING web server. The protein interaction network of other members of the PDF family is shown in the Fig. S5. The network nodes represent proteins. The network is divided into three clusters, represented by red, green and blue nodes. The edges represent the binding of proteins to proteins. The edges of different colors have different interactions. The purple edges represent the interactions between proteins, which has been experimentally proven. The black edges represent co-expression between proteins, and the yellow edges represent that they are still being explored.

Analysis of 2.0 kb genome sequences upstream of the *PDFs* start codon (ATG) showed that *PDFs* had rich *cis*-elements related to growth and development, hormone response, environmental stress, and light response. The identified environmental stress responsive *cis*-elements included ARE (anaerobic induced response element), DRE (drought response element), and LTR (low temperature response element). We also investigated the presence of ARR, CAAT-box, Dof, GATA box, TATA Box, WRKY, and MYB in the PDF promoter regions of both *B. napus* and *A. thaliana* (Fig. 5). As shown in Fig. 5, the results showed that DNA binding with one finger (Dof, AAAG) and Age-Related Resistance (ARR, GATT) were highly enriched, while WRKY and MYB binding sites were fewer in the *PDF* promoter regions.

The GO annotation results of *BnaPDFs* showed that they were mainly enriched in biological processes, such as killing of cells of other organism (GO:0031640), defense response (GO:0006952), and defense response to fungus (GO:0050832) (Fig. S7).

We predicted miRNAs targeting *BnaPDF* genes, and the results showed that the miRNAs only targeted *BnaA2.PDF2.5* and *BnaC7.PDF2.6*. The miRNAs including bna-miR164a, bna-miR164b, bna-miR164c, and bna-miR164d targeted *BnaA2.PDF2.5*, and these miRNAs exerted an inhibitory effect by directly cleaving mRNA. The miRNAs including bna-miR172a, bna-miR172d, and bna-miR6031 targeted *BnaC7.PDF2.6*. bna-miR172a, bna-miR172d inhibited the translation process, whereas bna-miR6031 directly cleaved mRNA (Table S1).

## Transcriptional analysis of *BnaPDFs* under diverse nutrient stresses

In order to explore potential biological functions of *BnaPDFs* in the growth and development of *B. napus*, the expression patterns of *BnaPDFs* were analyzed in different tissues, including blossomy pistil, bud ovule, leaf, new pistil, pericarp, root, sepal, silique, stamen, and stem (Fig. 6). It was observed that *BnaA6.PDF2.2* and *BnaC7.PDF2.2* were predominantly expressed in bud, silique, pericarp, and stamen, and *BnaA9.PDF2.3* was expressed at a high level in leaves, whereas *BnaA5.PDF1.2b1*, *BnaA5.PDF1.2b2*, and *BnaA5.PDF1.2b3* had the highest expression abundances in roots. However, the other *BnaPDF* members showed very low expression levels in various tissues.

In order to gain a deeper understanding of biological functions of *BnaPDFs* under abiotic stress, we analyzed the expression patterns of nitrate deficiency, ammonium toxicity, phosphorus deficiency, potassium deficiency, salt stress, and cadmium toxicity. Under nitrate deficiency, all the 37 *BnaPDFs* were not differentially expressed in the shoots, whereas seven DEGs were identified in the roots (Fig. 7). Most of the DEGs, particularly *BnaPDF2.2s*, *BnaPDF2.3s*, and *BnaPDF2.5s*, were obviously upregulated under nitrate deficiency (Figs. 7C–7G). However, the expression of *BnaC2.PDF1.2a2* and *BnaC6. PDF1.3* were suppressed in (Figs. 7A, 7B). Five members of *BnaPDFs* were differentially expressed under ammonium toxicity. It was noteworthy that the expression of *BnaPDF2.2s*, *BnaPDF2.3s*, and *BnaPDF2.5s* were induced in the shoots and roots under ammonium toxicity (Fig. 8).

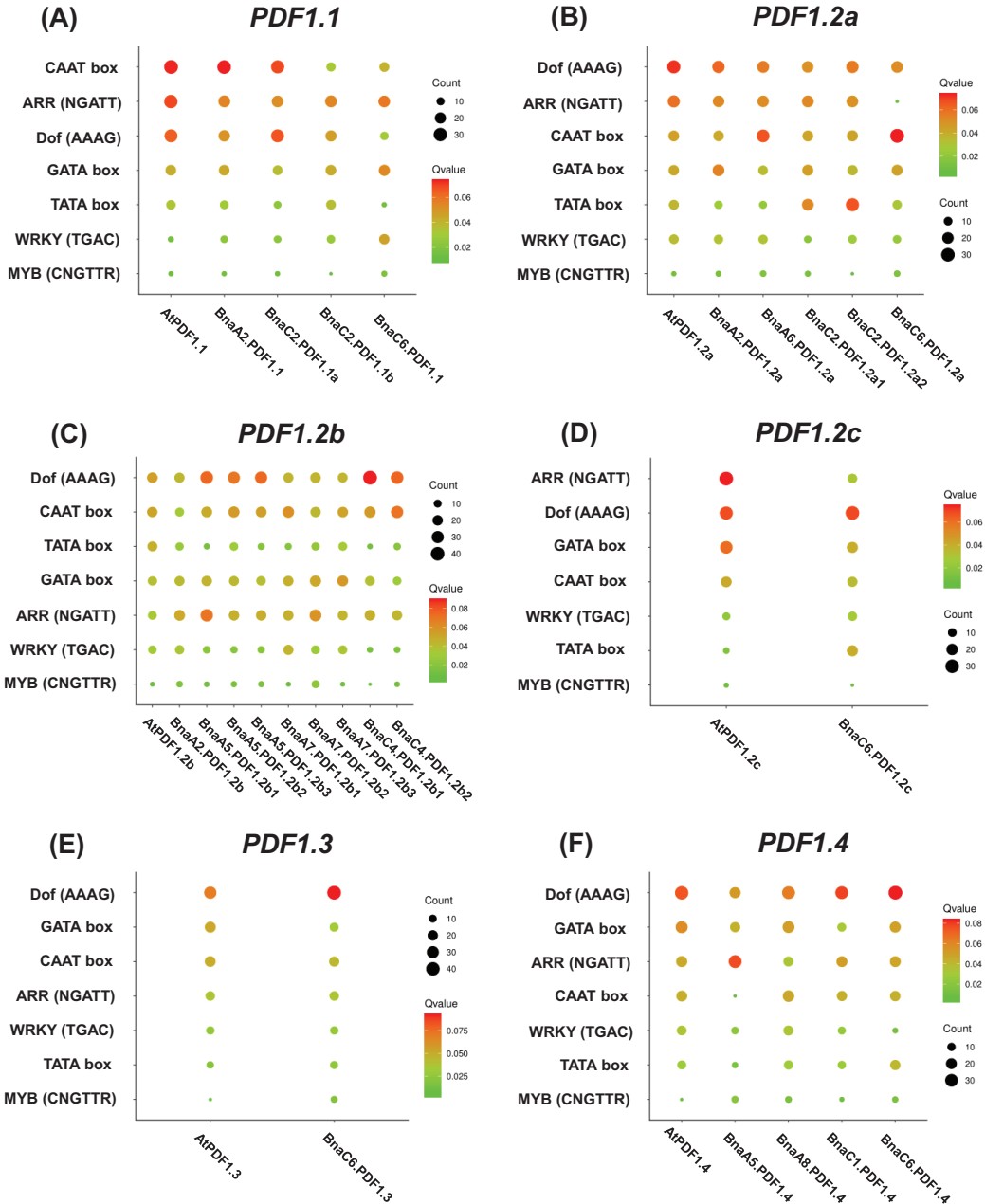

**Figure 5 Enrichment analysis of *cis*-acting regulatory elements (CRE) in the promoter region of *Brassica napus* plant defensins (PDF) genes.** Enrichment analysis of *cis*-acting regulatory elements of *PDF1.1* (A), *PDF1.2a* (B), *PDF1.2b* (C), *PDF1.2c* (D), *PDF1.3* (E), *PDF1.4* (F). the cis-acting elements of other members of the PDF family are shown in the Fig. S6. In the scatter chart, the larger the circle, the more CREs.

In general, the expression level of *BnaPDFs* was lower in the roots than in the shoots. Under low phosphate, we identified seven *BnaPDFs* that exhibited differential expression in shoots or roots, and a larger proportion of the DEGs were upregulated (Fig. 9).

Potassium deficiency leads to a decrease in the stomatal conductance of leaves and inhibition of photosynthesis. Potassium deficiency also reduces leaf expansion rate and leaf

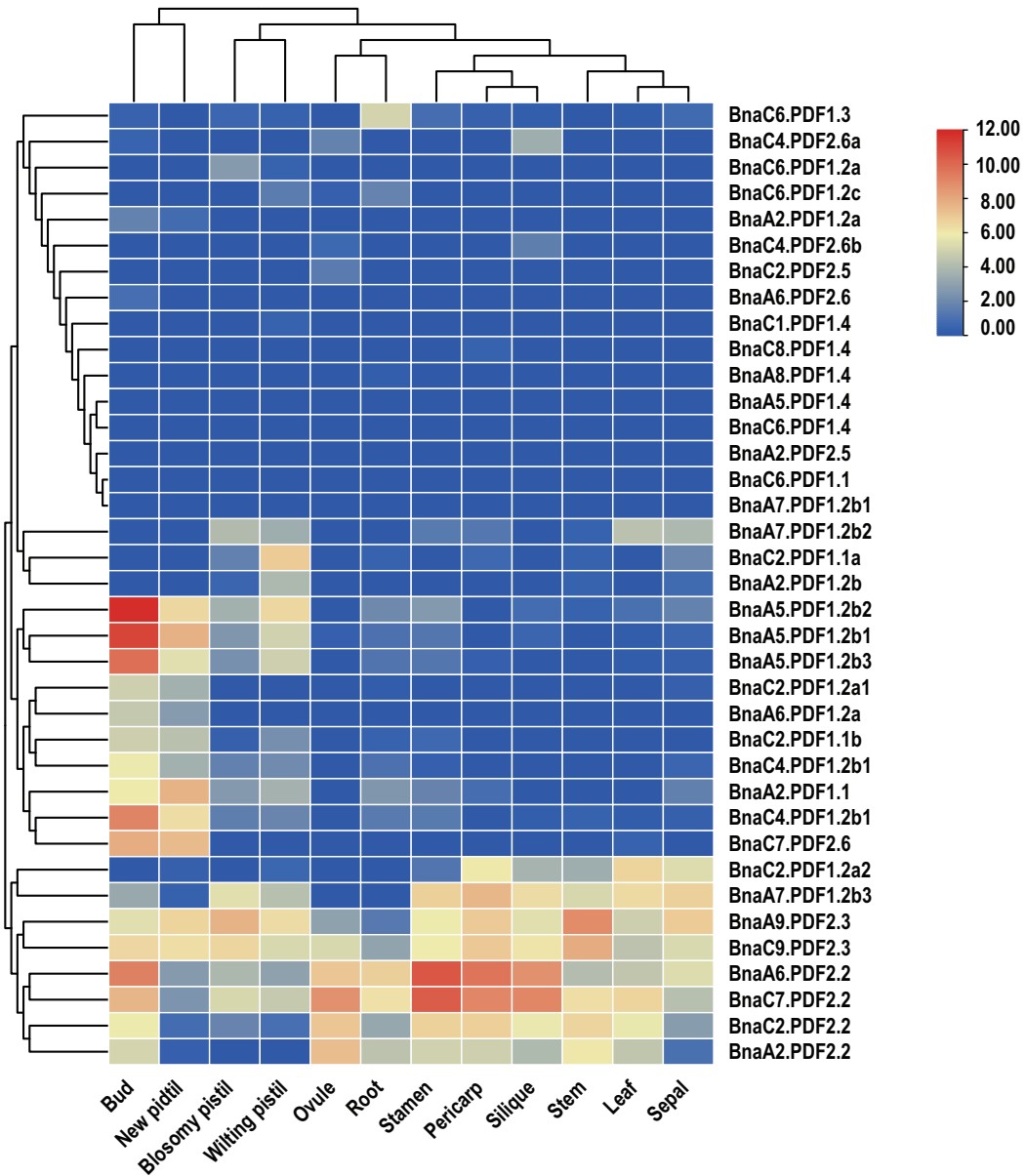

**Figure 6 Summarization of tissue/organ specificity expression of *PDFs* in *Brassica napus*.** Summarization of bud, new pistil, blosomy pistil, wilting pistil, ovule, root, silique, stamen, leaf, sepal, pericarp, stem specificity expression of *PDFs* in *Brassica napus*.

area (*Wang, Garvin & Kochian, 2002*; *Battie-Laclau et al., 2014*). Under potassium deficiency, a total of eight DEGs were identified in the shoots or roots (Fig. 10). In detail, *BnaC2.PDF1.2a2* and *BnaA7.PDF1.2b2* exhibited the highest levels in the shoots (Figs. 10A, 10B). In the shoots, five DEGs of *BnaPDF1.2b* and *BnaPDF1.2c* were upregulated under potassium deficiency (Figs. 10B–10D). In the roots, five downregulated (including *BnaPDF1.2s* and *BnaPDF2.2s*) and two up-regulated *BnaPDFs* (*BnaPDF1.4s*) were identified, respectively.

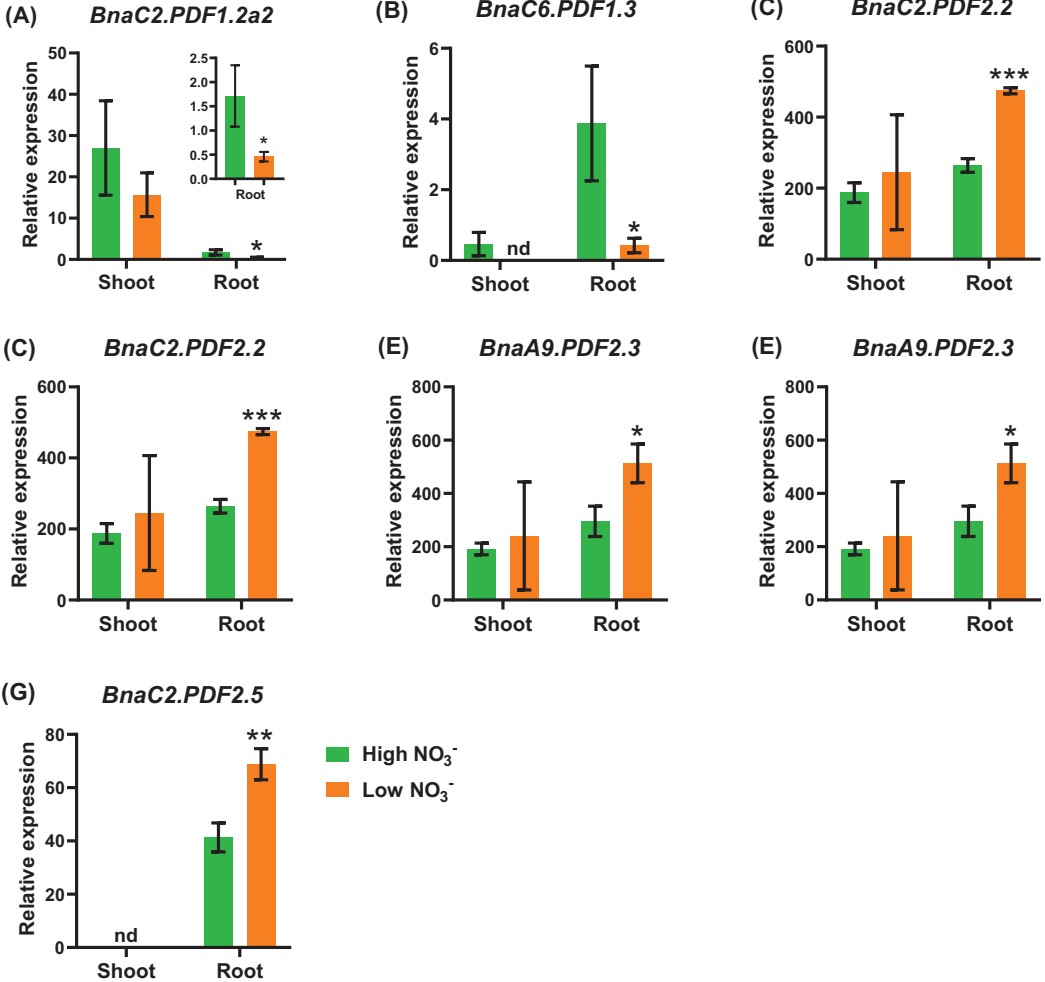

**Figure 7 The qRT-PCR-assisted transcriptional characterization of the *plant defensins* (PDF) genes in *Brassica napus* under different nitrate (NO$_3^-$) supply levels.** Differential expression of *BnaC2.PDF1.2a2* (A), *BnaC6.PDF1.3* (B), *BnaC2.PDF2.2* (C), *BnaC7.PDF2.2* (D), *BnaA9.PDF2.3* (E), *BnaC9.PDF2.3* (F), and *BnaC2.PDF2.5* (G) under high and low nitrate conditions. For high-throughput transcript profiling, the 7-day-old uniform *B. napus* seedlings after seed germination were hydroponic cultured in 6.0 mM nitrate for 10 d, and then were transferred to 0.30 mM nitrate for 3 d until sampling. The shoots and roots are three independent biological replicates. Error bars indicate SD ($n = 3$). nd stands for "not detected". An asterisk indicates that the *BnaPDFs* differentially expressed are significant at $^*P < 0.05$, $^{**}P < 0.01$, $^{***}P < 0.001$.

*B. napus* was a moderate-salt-tolerant crop and can be used as a phytoremediation resource for saline-alkali soils. In the shoots and roots, a total of eight *BnaPDF* DEGs were identified under salt stress. Under salt stress, *BnaA7.PDF1.2b3* had the highest expression level in the shoots, and *BnaA9.PDF2.3* had the highest expression level in the roots (Figs. 11B, 11E). The shoots upregulated genes included *BnaA7.PDF1.2b3*, *BnaC6.PDF1.3*, and *BnaC9.PDF2.3*, The roots upregulated genes included *BnaA7.PDF1.2b3*, *BnaA2.PDF2.5*, and *BnaC2.PDF2.5* (Fig. 11).

Cadmium, a heavy metal with strong biotoxicity, which can cause plant tissue cells to produce reactive oxygen species, cause membrane lipid peroxidation, and inhibit plant

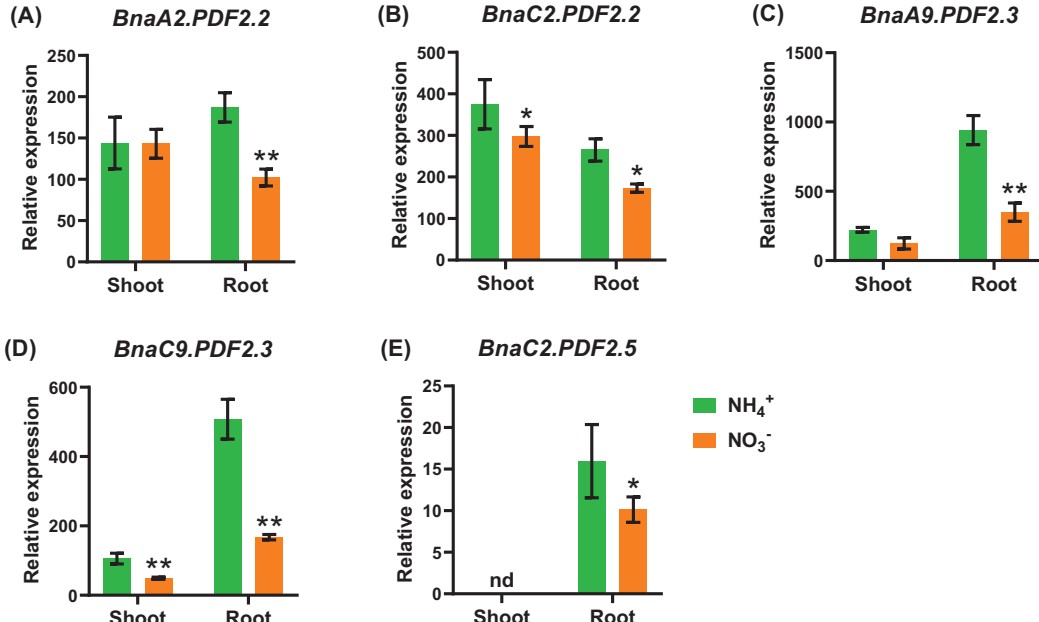

**Figure 8 The qRT-PCR-assisted transcriptional characterization of the *plant defensins* (PDF) in *Brassica napus* under different nitrogen (N) form conditions.** Differential expression of *BnaA2. PDF2.2* (A), *BnaC2.PDF2.2* (B), *BnaA9.PDF2.3* (C), *BnaC9.PDF2.3* (D), and *BnaC2.PDF2.5* (E) under nitrate ($NO_3^-$) and ammonium ($NH_4^+$) conditions. For high-throughput transcript profiling, the 7-day-old uniform *B. napus* seedlings after seed germination were hydroponic cultured in 6.0 mM nitrate for 10d, and then were transferred to an N-free solution for 3 d. Subsequently, seedlings were treated with 6.0 mM ammonium ($NH_4^+$) for 3 d until sampling. The shoots and roots are three independent biological replicates. Error bars indicate SD ($n = 3$). nd stands for "not detected". An asterisk indicates that the *BnaPDFs* differentially expressed are significant at $^*P < 0.05$, $^{**}P < 0.01$.

growth (*Pereira de Araújo et al., 2017*; *Ben Ghnaya et al., 2009*). We identified a total of ten *BnaPDF* DEGs in the shoots and roots under cadmium toxicity (Fig. 12). All the expression of *BnaC2.PDF1.2a2*, *BnaC2.PDF2.2*, and *BnaC7.PDF2.2* was induced by cadmium in the shoots and roots (Figs. 12A, 12G, 12H). In the shoots, we found that the cadmium toxicity significantly induced the expression of the *BnaPDF1.2a*, *BnaPDF1.2b*, and *BnaPDF2.2* DEGs, whereas repressed the expression of the *BnaC7.PDF2.2*. In the roots, the expression of most *BnaPDFs* was significantly downregulated by cadmium toxicity except that the *BnaC6.PDF1.4* expression was obviously upregulated by cadmium toxicity.

According to the transcriptional responses of *BnaPDFs* to multiple nutrient stresses, we conducted a summary analysis. These results showed that all the *BnaPDFs* were not identified to responsive to the six nutrient stresses above-mentioned, whereas *BnaC2. PDF1.2a2* was simultaneously responsive to five nutrient stresses, including low nitrate, limited phosphate, potassium deficiency, toxic cadmium and salt stress. In addition, *BnaA9.PDF2.3*, *BnaC9.PDF2.3*, *BnaA2.PDF2.2*, *BnaA7.PDF1.2b3*, *BnaC7.PDF2.2*, *BnaC2. PDF2.2*, and *BnaC6.PDF1.3* were simultaneously responsive to three or four nutritional stresses (Table 4).

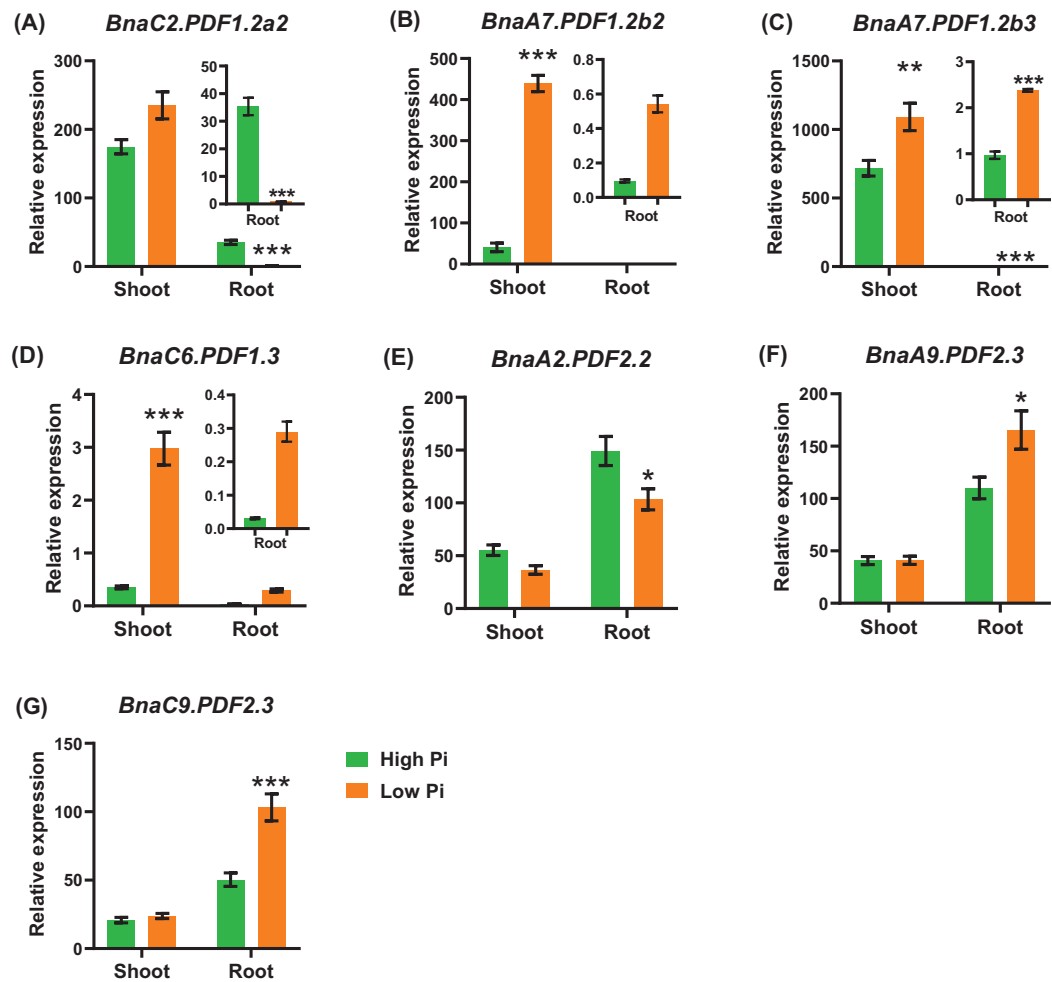

**Figure 9 The qRT-PCR-assisted transcriptional characterization of the *plant defensins* (*PDFs*) in *Brassica napus* under different phosphate (Pi) levels.** Differential expression of *BnaC2.PDF1.2a2* (A), *BnaA7.PDF1.2b2* (B), *BnaA7.PDF1.2b3* (C), *BnaC6.PDF1.3* (D), *BnaA2.PDF2.2* (E), *BnaA9.PDF2.3* (F), and *BnaC9.PDF2.3* (G) under high Pi and low Pi levels. For high-throughput transcript profiling, the 7-day-old uniform *B. napus* seedlings after seed germination were hydroponic cultured in 250 μM Pi for 10 d, and then were transferred to 5 μM Pi for 3 d until sampling. The shoots and roots are three independent biological replicates. Error bars indicate SD (*n* = 3). An asterisk indicates that the *BnaPDFs* differentially expressed are significant at *P < 0.05, **P < 0.01, ***P < 0.001.

# DISCUSSION

In the process of gene evolution, conserved regions are retained, but several mutations have also occurred, along with changes in gene function, leading to the differentiation of genes into subfamily (*Ohno, 1999*). *B. napus* is an allotetraploid plant species formed by natural hybridization of two diploids, *B. rapa* and *B. oleracea*, and undergoes several rounds of whole-genome triplication and duplication compared with Arabidopsis (*Bayer et al., 2017*). Therefore, these processes usually result in the formation of multicopy gene families in *B. napus* (*Parkin et al., 2005*). Previous studies have shown that the existence of defensins is universal, small in size (45–54 amino acid residues), relative

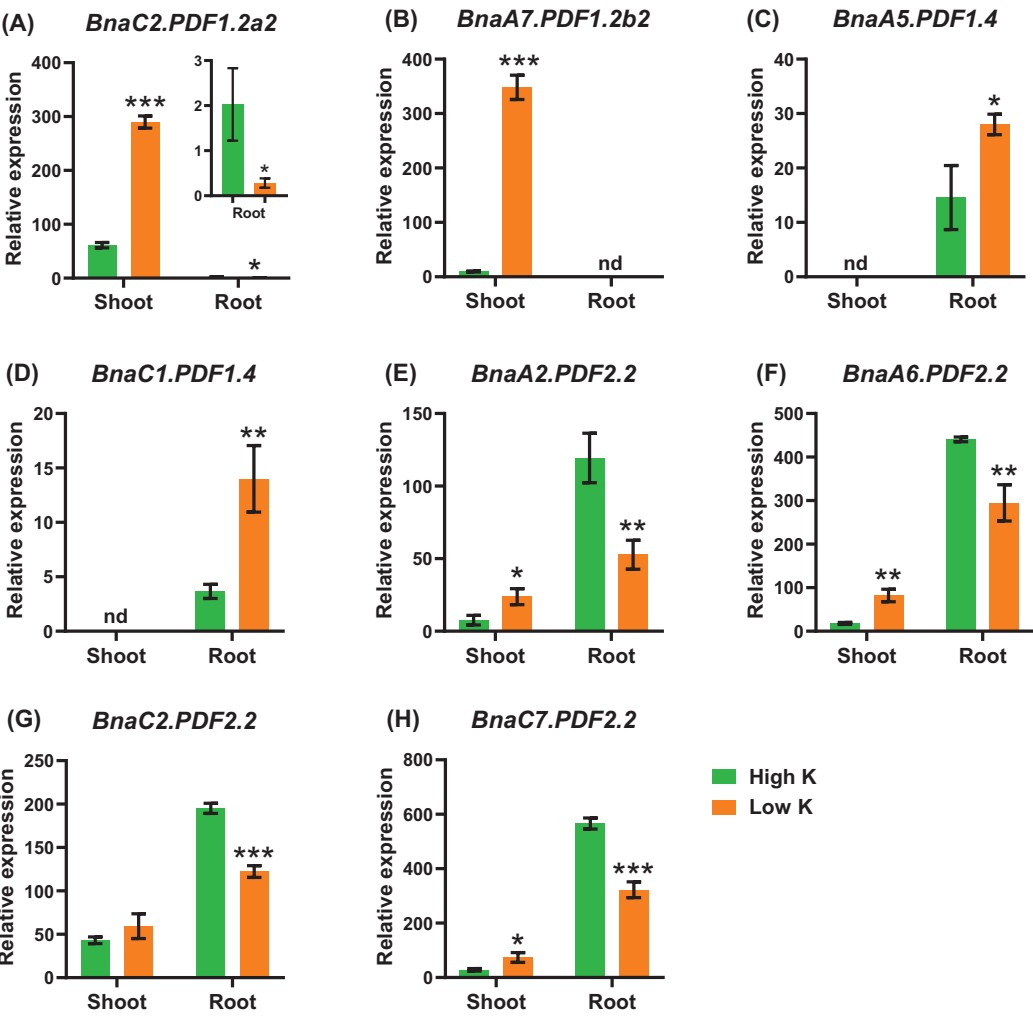

**Figure 10 The qRT-PCR-assisted transcriptional characterization of the *plant defensins* (PDFs) in *Brassica napus* under different potassium (K) levels.** Differential expression of *BnaC2.PDF1.2a2* (A), *BnaA7.PDF1.2b2* (B), *BnaA7.PDF1.2b3* (C), *BnaC6.PDF1.3* (D), *BnaA2.PDF2.2* (E), *BnaA9.PDF2.3* (F), and *BnaC9.PDF2.3* (G) under high K and low K levels. For the transcriptional analysis, the 7 d-old uniform *B. napus* seedlings after seed germination were first hydroponically grown under 6 mM $K^+$ 10 d, and then were transferred to 0.03 mM $K^+$ for 1 d until sampling. The shoots and roots are three independent biological replicates. Error bars indicate SD ($n = 3$). nd stands for "not detected". An asterisk indicates that the *BnaPDFs* differentially expressed are significant at $^*P < 0.05$, $^{**}P < 0.01$, $^{***}P < 0.001$.

molecular mass (5–7 kDa). The defensins usually contain eight cysteines forming four disulfide bonds, forming three antiparallel β-chains and one α-helix in a conservative conformation, which is considered necessary for structure and function (*Cools et al., 2017a*; *Campos et al., 2018*; *Bhattacharya et al., 2017*). However, there have been few systematic studies on *PDFs* in *B. napus* so far. The genome-wide identification of *BnaPDFs* will provide a comprehensive insight into their family evolution and provide elite gene resources for the genetic improvement of rapeseed resistance to nutrient stresses.

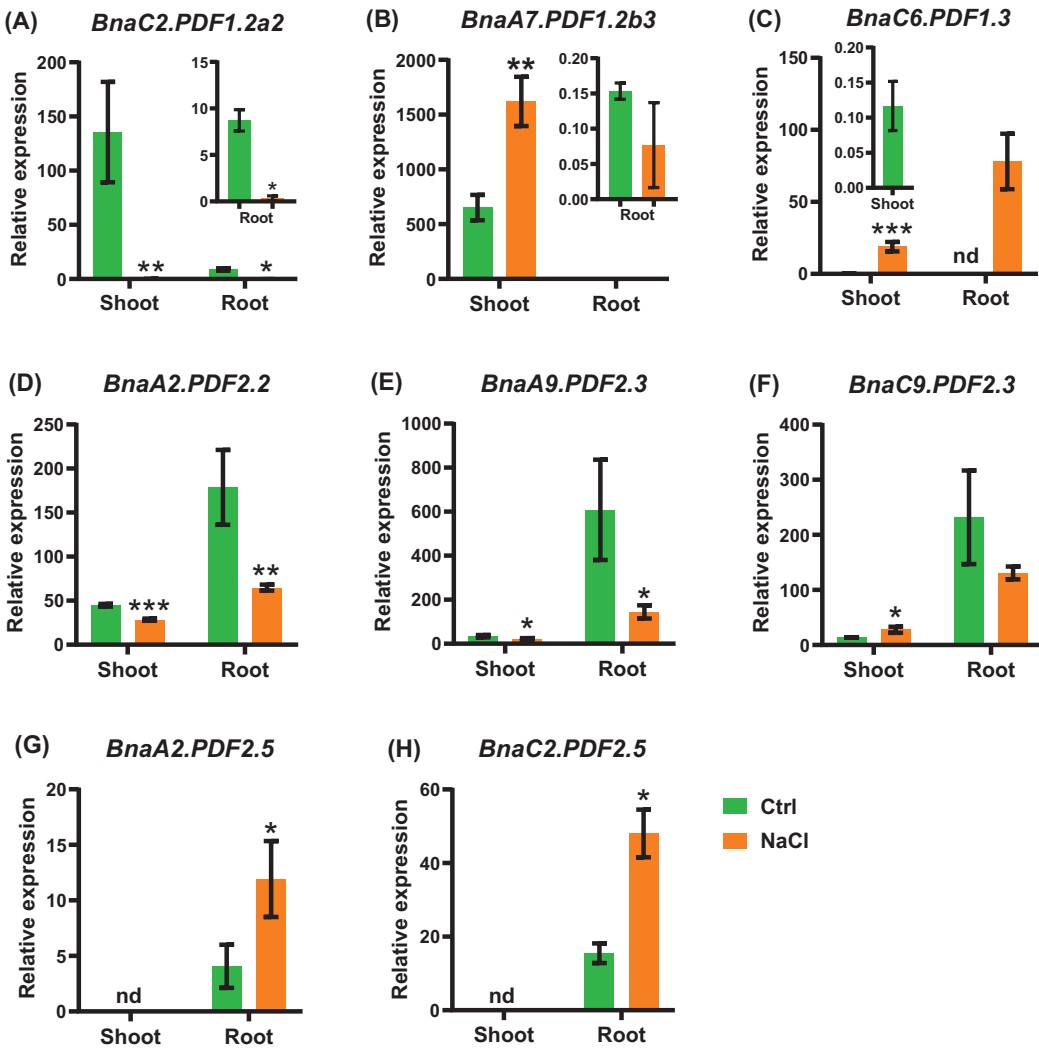

**Figure 11 The qRT-PCR assisted transcriptional characterization of the *plant defensins* (PDFs) in *Brassica napus* under salt stress.** (A–H) Differential expression profiling of: *BnaC2.PDF1.2a2* (A), *BnaA7.PDF1.2b3* (B), *BnaC6.PDF1.3* (C), *BnaA2.PDF2.2* (D), *BnaA9.PDF2.3* (E), *BnaC9.PDF2.3* (F), *BnaA2.PDF2.5* (G), and *BnaC2.PDF2.5* (H) under salt stress. For high-throughput transcript profiling, the 7-day-old uniform *B. napus* seedlings after seed germination were hydroponic cultured in NaCl-free solution for 10 d, and then transferred to 200 mM NaCl for 12 h until sampling. The shoots and roots are three independent biological replicates. Error bars indicate SD ($n = 3$). nd stands for "not detected". An asterisk indicates that the *BnaPDFs* differentially expressed are significant at *$P < 0.05$, **$P < 0.01$, ***$P < 0.001$.

## Comprehensive analysis of molecular characteristics of BnaPDFs

The secondary structures of BnaPDFs were predicted to contain α-helix, extended chain, β-turn, and random coils, of which α-helix and random coils were the main components (Fig. S4). Most members contained 1–3 exons, and the gene structure was relatively simple.

Plant defensins can be divided into two classes according to their precursors (*Lay & Anderson, 2005*). Class I defensins composed of signal peptides and mature defensins and secreted into the extracellular space (*Parisi et al., 2019*). Class II defensins contained a C-terminal propeptide, and is mainly expressed constitutively in flowers and fruits of

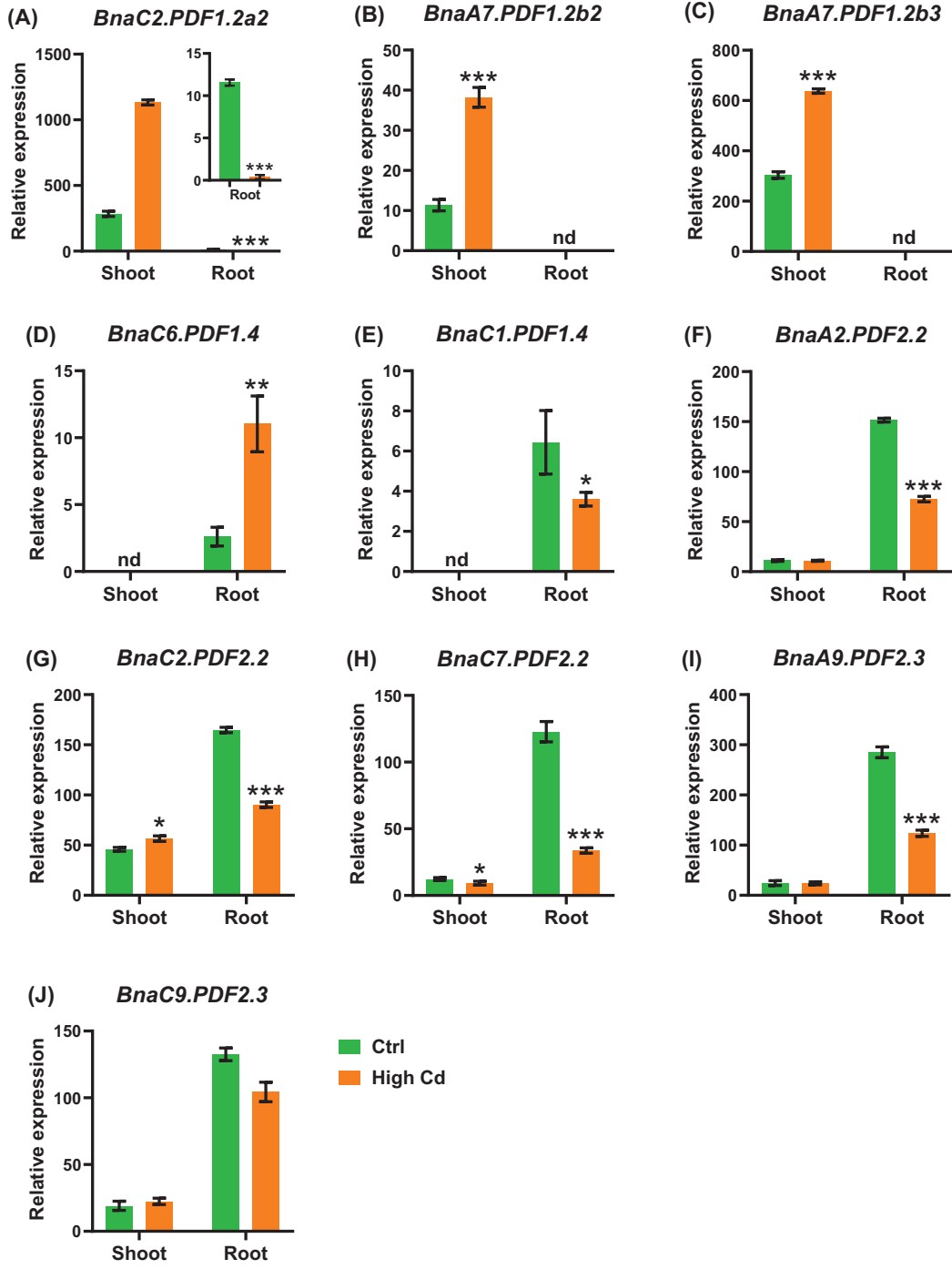

**Figure 12** **The qRT-PCR-assisted transcriptional characterization of the *plant defensins* (PDF) genes in *Brassica napus* under cadmium (Cd) toxicity.** Differential expression of *BnaC2.PDF1.2a2* (A), *BnaA7.PDF1.2b2* (B), *BnaA7.PDF1.2b3* (C), *BnaC1.PDF1.4* (D), *BnaC6.PDF1.4* (E), *BnaA2.PDF2.2* (F), *BnaC2.PDF2.2* (G), *BnaC7.PDF2.2* (H), *BnaA9.PDF2.3* (I), and *BnaC9.PDF2.3* (J) under Cd-free (−Cd) and Cd (10 μM CdCl$_2$) toxicity. For the transcriptional analysis, the 7-d-old uniform *B. napus* seedlings after seed germination were hydroponically cultivated in a cadmium-free solution for 10 d, and then were transferred to 10 μM CdCl$_2$ for 12 h until sampling. The shoots and roots are three independent biological replicates. Error bars indicate SD ($n = 3$). nd stands for "not detected". An asterisk indicates that the *BnaPDFs* differentially expressed are significant at *$P < 0.05$, **$P < 0.01$, ***$P < 0.001$.

**Table 4 Number of differentially expressed *PDFs* under multiple nutrient stresses.**

| Term | Nitrate limitation | Ammonium excess | Phosphorus starvation | Potassium deficiency | Cadmium toxicity | Salt stress |
|------|--------------------|-----------------|-----------------------|----------------------|------------------|-------------|
| *PDF1s* | 2 | 3 | 3 | 4 | 5 | 3 |
| *PDF2s* | 5 | 6 | 4 | 4 | 5 | 5 |

Solanaceae plants (*Balandín et al., 2005*). We used WoLF PSORT to subcellular localization of the PDF gene, and the result showed that it is extracellular region or secreted. Our predicted results are consistent with the characteristics of class I of defensins. In addition, all BnaPDF members contain signal peptides. Previous studies have confirmed that the signal peptide guides the newly synthesized protein to be secreted outside the organelles to perform functions (*Hiller et al., 2004*). Therefore, BnaPDF is synthesized in the cell and functions in the extracellular region.

Based on the protein-protein interaction network, we found that PDFs may interact with ethylene-responsive transcription factors, disease-related proteins, 1,3-glucanase and chitinase. Plants produced ethylene response transcription factors to reduce the attraction of herbivorous pests, and also activated the downstream transcriptional response of jasmonic acid, conferring resistance to several necrotrophic fungi (*Berrocal-Lobo, Molina & Solano, 2002*; *Miyamoto et al., 2019*). PDFs were also related to 1,3-glucanase and chitinase. Glucan and chitin were the main components of fungal cells, therefore, PDFs had strong antibacterial activity (*Tian et al., 2007*). According to the evolutionary relationship, most gene pairs had short branch lengths, indicating that differentiation has occurred recently.

Transcription factors initiated the transcription process of specific genes by interacting with *cis*-acting elements, thereby regulating gene expression (*Wittkopp & Kalay, 2011*). In this study, a variety of plant hormones and stress-related *cis*-elements were identified in the promoter regions of the *BnaPDF* family members, indicating that *BnaPDFs* might play an essential role in the adaption of rapeseed plants to stresses. miRNA is an important participant in mediating plant immunity to biological stress. It enhances the plant immune system by regulating the expression of plant hormones and target genes (*Fei et al., 2016*). At present, there are few researches on the interaction of PDFs and miRNAs, but this field should be studied more deeply. Several bioinformatic tools were employed to predict miRNAs upstream of PDFs in the study, and this approach may provide a new perspective for understanding plant defense mechanisms. It has been reported that miR-124 and miR-924 are involved in the pathogenesis of inflammatory bowel disease by negatively regulating α-defensin 5 mRNA and protein expression (*Miles et al., 2016*).

## Potential involvement of BnaPDFs in rapeseed responses to diverse nutrient stresses

Plant defense response regulates the growth and development of plant roots, leaves, stems, bud, new pistil, blossomy pistil, ovule, silique, stamen, and sepal (*Hegedüs & Marx, 2013*). Tomato *SlDef2* regulates flower development. Overexpression *SlDef2* reduces pollen
viability and seed production (*Stotz, Spence & Wang, 2009*). In this study, *BnaA6.PDF2.2* and *BnaC7.PDF2.2* were identified to be predominantly expressed in bud, silique, and stamen. However, *BnaA5.PDF1.2b1*, *BnaA5.PDF1.2b2*, and *BnaA5.PDF1.2b3* had the highest expression levels in the bud, showed that different *PDF* members might play differential roles in different parts of rapeseed plants. It also implied that *PDFs* might be involved in the growth and development of the leaf, root, stamen, stem, and other organs of *B. napus*.

Nitrogen is an essential nutrient for plants, nitrate and ammonium are the two primary inorganic nitrogen sources absorbed and utilized by plants (*Konishi & Yanagisawa, 2014*). After plants absorb nitrate, part of it is directly transported to shoots or stored in vacuoles. The other part is transported to the shoot and converted to nitrite under nitrate reductase in the leaves. Nitrite reductase produces ammonium, which is assimilated into amino acids by the GS/GOGAT pathway (*Tischner, 2000*; *Clément et al., 2018*). Nitrogen absorption is a key step in nitrogen metabolism, and is one of the most important limiting factors for plant growth. Nitrogen deficiency seriously affects plants' growth and development. Therefore, improving nitrogen use efficiency (NUE) is an important aspect of improving plant resistance to nitrogen deficiency (*Hammad et al., 2017*). This study analyzed the expression of *BnaPDF* family members under nitrate deficiency and ammonium toxicity treatments. Under nitrogen deficiency, chloroplast proteins in senescent leaves are degraded into amide-nitrogen compounds and redistributed to newly developed leaves (*Hammad et al., 2017*).We found that the general expression of *PDF1.2s*, *PDF2.2s*, *PDF2.3s*, and *PDF2.5* was induced by nitrate limitation (Fig. 7). Several previous studies have shown that small peptides can be used as signal molecules in the protein kinase pathway to indirectly regulate the expression of other genes. *AtPDF2.1* mediates ammonia metabolism by regulating the activity of glutamine synthase in Arabidopsis (*Yao et al., 2019*). When plants are exposed to ammonium stresses, *PDF* might regulate the activity of glutamine synthetase activity, thereby affecting the concentration of glutamine and the movement of NADH-GOGAT (*Yao et al., 2019*).

Phosphorus is one of indispensable macronutrient elements in the process of plant growth and development. It is an important part of many metabolites and macromolecules such as ATP, phospholipids, and nucleic acids in plants (*Wang, Garvin & Kochian, 2002*; *Hu et al., 2019*). In this study, most *BnaPDFs* are upregulated under phosphate starvation (Fig. 9), which indicated that *BnaPDFs* might play important roles in the resistance of rapeseed plants to phosphate deficiency. OsAFP1 interacts with phosphate ion, this ion is fixed by OsAFP1 dimer interface, the side chains of His37 in OsAFP1 coordinate with phosphate ion (*Ochiai et al., 2020*). HsAFP1 interacts with the phosphate group in phosphatidylinositol phosphate to induce changes in membrane permeability and exert antifungal activity (*Cools et al., 2017b*).

Potassium plays an important role in various physiological and metabolic processes, such as cell osmotic regulation, enzyme activation, chlorophyll synthesis, stomatal movement, and signal transduction (*Wang, Garvin & Kochian, 2002*). Under potassium deficiency, it was noted that the expression of most *BnaPDF* DEGs was significantly induced in the shoots but was inhibited in the roots (Fig. 10). Plant defensins are used as

potassium channel blockers (*Parisi et al., 2019*; *Vriens et al., 2016*). For example, the sequence of *AtPDF2.3* contains a toxin characteristic sequence (K-C5-R-G) that can block potassium channels (*Vriens et al., 2016*). The defensin-like ZmES4 is exclusively expressed in female gametes and interacts with the potassium channel KZM1, further leading to potassium influx (*Amien et al., 2010*). We also found the same toxin feature (K-C5-RG) in BnaPDF2.3s, so we speculate that BnaPDF2.3s and potassium channels exist interaction. We speculated that *BnaPDFs* might be involved in regulating the ion balance in cells and maintaining plant cell homeostasis (*Fisher et al., 2012*).

Soil salinity is one of the important environmental factors that restrict plant growth and development, and salt stress significantly reduces rapeseed yield (*Munns & Tester, 2008*; *Ashraf & McNeilly, 2004*). The expression of most *BnaPDF* DEGs was induced by salt stress (Fig. 11), and they might improve the salt tolerance of rapeseed plants. Indeed, *PDFs* were reported to be upregulated in salt stress, which activate the ROS scavenging system in plant cells, thereby protecting the photosystem and enhancing plant tolerance to abiotic stress (*Khadka et al., 2020*; *Wu, Lin & Chuang, 2016*).

Plant defensins chelate with cadmium through sulfhydryl groups, promote the secretion of cadmium outside the cell, reduce the intracellular cadmium content, and participate in cadmium transport and distribution (*Luo et al., 2019b*; *Ben Ghnaya et al., 2009*). *AtPDF2.5* is significantly induced under cadmium stress. *AtPDF2.5* reduces the free cadmium in the cell and enhance the resistance of Arabidopsis to cadmium by chelating the intracellular cadmium and mediating its efflux (*Luo et al., 2019b*). *AtPDF2.6* also improves plant tolerance to cadmium by chelating with cadmium (*Luo et al., 2019a*). In this study, we found that the expression of some *BnaPDFs* was altered in response to cadmium toxicity, which indicated their potential participation in alleviating the toxicity of cadmium to rapeseed plants, and further promoted the combination of cadmium-chelators. In summary, analysis of tissue-specificity expression patterns and the transcriptional responses of *BnaPDFs* to various nutrient stresses indicated that *BnaPDFs* might play an essential role in different growth stages and stress responses of *B. napus*.

Plant diseases, especially fungal diseases, are one of the main reasons for crop yield reduction. At present, the application of plant defensins with broad-spectrum antibacterial activity in crop disease-resistant breeding is called a hot spot (*Fisher et al., 2012*). After the alfalfa defensin, *MsDef1* was expressed in potatoes, and potatoes showed strong resistance to *Verticillium Dahliae* (*Gao et al., 2000*). The expression of alfalfa defensin *MsDef4.2* in wheat enhances wheat resistance to leaf rust but does not affect the colonization of beneficial arbuscular mycorrhizal fungi on the roots (*Kaur et al., 2017*). The co-expression of *Rs-AFP1* and chimeric chitinase in rapeseed enhances the resistance of *Brassica napus* to *Sclerotinia sclerotiorum*.

## CONCLUSION

In this study, based on the identification and molecular characterization of the *PDFs*, the *BnaPDF* DEGs that were identified under various nutrient stresses might be used as elite gene resources for the genetic improvement of rapeseed plants to stresses. In addition, the research results could also provide references for a deeper understanding of the

molecular evolution mechanism and potential functions of the *PDF* family members in *B. napus.*

## ACKNOWLEDGEMENTS

We thank all the colleagues in our laboratory for providing useful discussions and technical assistance. We are very grateful to the editor and reviewers for critically evaluating the manuscript and providing constructive comments for its improvement.

## ABBREVIATIONS

| | |
|---|---|
| **AMP** | antimicrobial peptides |
| **BRAD** | Brassica Database |
| **CDSs** | coding sequences |
| **CREs** | cis-acting regulatory elements |
| **gDNA** | genomic DNA |
| **GOGAT** | glutamine oxoglutarate aminotransferase |
| **GRAVY** | grand average of hydropathicity |
| **GS** | glutamine synthetase |
| **N** | Nitrogen |
| **Ks** | the synonymous substitution rate |
| **Ka** | non-synonymous substitution rate |
| **NUE** | N use efficiency |
| **PDF** | plant defensins |
| **pI** | isoelectric point |
| **STRING** | Search Tool for Recurring Instances of Neighboring Genes |
| **TAIR** | The Arabidopsis Information Resource |
| **II** | instability index. |

### Funding

This study was financially supported by the National Natural Science Foundation of China (31801923), Major Collaborative Innovation Project of Zhengzhou City (Key Discipline Construction Project of Zhengzhou University) (NO. xkzdjc201905), and the Youth Innovation Project of Key discipline of Zhengzhou University (NO. XKZDQN202002). The funders had no role in study design, data collection and analysis, decision to publish, or preparation of the manuscript.

### Grant Disclosures

The following grant information was disclosed by the authors:
National Natural Science Foundation of China: 31801923.
Major Collaborative Innovation Project of Zhengzhou City: xkzdjc201905.
Youth Innovation Project of Key discipline of Zhengzhou University: XKZDQN202002.

## Competing Interests

The authors declare that they have no competing interests.

## Author Contributions

- Ying Liu performed the experiments, analyzed the data, prepared figures and/or tables, and approved the final draft.
- Ying-peng Hua conceived and designed the experiments, analyzed the data, authored or reviewed drafts of the paper, and approved the final draft.
- Huan Chen analyzed the data, authored or reviewed drafts of the paper, and approved the final draft.
- Ting Zhou analyzed the data, authored or reviewed drafts of the paper, and approved the final draft.
- Cai-peng Yue analyzed the data, authored or reviewed drafts of the paper, and approved the final draft.
- Jin-yong Huang conceived and designed the experiments, authored or reviewed drafts of the paper, and approved the final draft.

## Data Availability

Raw measurements are available in the Supplemental Files.

## Supplemental Information

Supplemental information for this article can be found online at http://dx.doi.org/10.7717/peerj.12007#supplemental-information.

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
