# Peer review of "Genome-scale identification of *plant defensin* (*PDF*) family genes and molecular characterization of their responses to diverse nutrient stresses in allotetraploid rapeseed"

_PeerJ, doi:10.7717/peerj.12007_

## Round 0.1 · original submission · Major Revisions

My suggested changes and reviewer comments are shown below and on your article 'Overview' screen.

Reviewer 1 ·

Basic reporting

no comment

Experimental design

no comment

Validity of the findings

no comment

Additional comments

The manuscript entitled “Genome-scale identification of plant defensin (PDF) family genes and molecular characterization of their responses to diverse nutrient stresses in allotetraploid rapeseed” submitted by Liu et al shows great effort in elucidating plant defensin gene divergence in allotetraploid rapeseed and how these genes expression change in response to diverse nutrient stresses. This work is very important for the in-depth study of PDF function. However, several content organization need to be slightly rephrased to highlight importance of their work before it can be accepted for publication.

1. Line 26: change into following should be clear. “Three distinct clades were identified in BnaPDfs phylogeny. Clade specific conserved motifs were identified within each clade respectively.” Please check your MEME results and denote those domains.

2. Line 120-125: Please add more detail and explain how it work briefly.

3. Line 142: format problem, please check.

4. Line 217: “The shoots and roots were sampled separately and were…” What kinds of shoots and roots?? Please make it clear.

5. Line 293-295, 299-300, 304-305, 315-317, 392-394, 498-400, 408-409: please move to introduction.

6. Gene structure of your target gene exhibits great variety. Please check your Figure 4. The intron and exon length differs among your target genes.

7. Line 459-467: Here is a repeat of your results, not discussion. Please rephrase.

8. Line 510-511: This is rough. Please make your statement clear.

9. The following articles about the function of defensins CAL2, AtPDF2.2 and AtPDF2.3 should be cited in the background.
(1) Luo JS, Xiao Y, Yao JY, Wu ZM, Yang Y, Ismail AM and Zhang ZH. Overexpression of a Defensin-Like Gene CAL2 Enhances Cadmium Accumulation in Plants. Frontiers in Plant Science. 2020, 11:217.
(2) Luo JS, Zhang ZH. Proteomic changes in the xylem sap of Brassica napus under cadmium stress and functional validation. BMC Plant Biology. 2019,19:280.

Reviewer 2 ·

Basic reporting

The manuscript submitted by Liu et al deal with the description of rather unknown family of proteins, defensins. This genome wide analysis is well performed, although I have several comments on the works.

The introduction have some spelling or English problems, as in line 50 and in line 72 (better to say Arg 38, is more scientific spelling). But my major problem is that the introduction needs more work about the role of defensins related to the experiment that are performed in the article. Authors perform experiments on nitrate, and other compounds that are not defined in the short introduction of this experiment.

Figures are extremely small, especially those that contain trees or domains (1, 2) and are almost impossible to follow.

The main fail of this paper is that many figures in the discussion present data that are impossible to see, like 11 A, B, D. 12 B, C (this even have a negativa caption in the axis). And there is statistic asterisk that doesn't make any sense. In every footnote, authors only claim that asterisk indicate a P<0.05 statistic difference but in the figures you include, indistinctly one, two or three stars, without indicate what that does mean!. Authors should improve the figures, like breaking the axes to star reviewing this data,

Experimental design

For the identification of brassica PDF I don't understand how the authors perform that. I don't get if the identification has been made only with the PDF domain or if the whole A.thaliana proteins have been used. Moreover, authors perform a whole identification of PDF in A. thaliana (even the chromosome identification), when this is not the scope of this work.

In the line 124 authors claim that if a gene is within a 100kb region are defined as tandem. This is a very large distance, and did not present any reference for that. Also, have you checked that this putative tandem genes are similar between them?
The reference used in line 228 about the AACt calculation is not accurate, as is not the one that explain that methods.

When authors performed phylogenetic tree, like in the figure 2B, they should include the bootstrap number moreover because they say they calculated It (line 133).

It is noteworthy that you haven't performed any curation on the sequences obtained. Other articles on genome-wide investigation have discovered genes that there is genes that may contain not complete domains or duplication that needed to be curated. Did you find some genes like that?

Validity of the findings

Between lines 245 and 250, author preform a identification of A.thaliana PDF, which is not the scope of this work.

In lines 261-263 authors claim that those genes are tandem, are those genes more similar between them than others located in other Chr?

When author perform MEME to identify domains in the sequence, have you checked if any of those domains are predicted in proteins databases as PDB?

In the results section, when describing the qPCR data, you perform an introduction, like in lines 377 to 383 that are more a discussion rather than results. You should move these description to the discussion.

In line 455, how can be the extracellular matrix the site of PDF biosynthesis?

Lines 463 and 465 why are this lines here? What do you want to say with that? It the same with the lines 468 to 471, you need to discuss more this section

In lines 505 to 509 you made a nice study of PDF related to potassium channels. Have you find any of this sequences in the brassica PDF sequences related with potassium?

As stated in the basic reporting part, authors should improve the qPCR figures to continue the revision.

---

## Round 0.2 · Major Revisions

In the process of reviewing manuscripts, two reviewers left, and I found another reviewer, but only one reviewer responded. I'm very sorry for the inconvenience.

Reviewer 3 ·

Basic reporting

no comment

Experimental design

no comment

Validity of the findings

no comment

Additional comments

Though, the authors have made a great effort in improving the manuscript. Still, I am not satisfied with the revised version. There is a scope for another round of revision. Please see the attached file without track changes for line numbering. Some of the comments are as follows:
 Line 27-28, please mention the common and scientific name together on the first appearance in the abstract and the main text, i.e., rapeseed (Brassica napus L.).
 Line 34, enriched elements. Please add enriched cis-elements, and the word “cis” should be in italic throughout the text.
 Line 37, please specify the tissues and specific genes.
 Please carefully check the entire text and make sure the genes name should be in italic, but not the protein.
 Line 97, Blackshaw et al., 2011 can be replaced with a recent review https://doi.org/10.1007/s00344-020-10231-z.
 Line 119-120, please mention the threshold e-value used to identify the AtPDFs using the Hidden Markov Model (HMM V 3.0) profiles.
 The legend is missing in Fig 1 (A and B).
 In table 1 and 2, please do not merge the Arabidopsis and rapeseed genes. Arrange them in order, firstly Bna followed by AtPDFs.
 In the methodology sections, there is no information about the statistical analysis used in Fig 7-12. Please add a separate section of “statistical data analysis”.
 Further, please rearrange the graphs (Fig 7-12). Remove the green and orange legend for each graph and keep it at the end near to the last graph, no need to repeat it for each graph. Further, there are so many extra spaces between the graphs. In one line, 3-4 graphs can easily be adjusted. So, please check and revise all graph figures.
 Did the authors try to predict the putative miRNA targeting BnPDF genes and GO annotation analysis? If no, I suggest adding it to the current version to add additional insights into the current work. This work will not take much time based on my own experience but greatly improved the work strength. For miRNA prediction, the authors can try with psRNATarget database http://plantgrn.noble.org/psRNATarget/. If you can identify any miRNA targeting any of your genes, you can add the results accordingly. If there are no miRNAs, please omit them. However, you can try GO annotation results.
 Line 460, 476, this paragraph is just the repetition of the results part. Please try to compare your findings with the previous ones and support your results.
 Line 591-595, make a separate conclusion section and update the conclusion with future directions.
 The English language needs improvement.

Annotated reviews are not available for download in order to protect the identity of reviewers who chose to remain anonymous.

---

## Round 0.3 · accepted · Accept

Sorry to delay you for such a long time, this manuscript review process is also very bumpy, but the final result is good, thank you for holding on until the manuscript review is over!

The Section Editor asked that you address the following items in the production process:

1. I think the additional comments from Reviewer 3 regarding some of the figures and supplementary information should be addressed - they haven't yet according to what I can see in the current submission files.

2. Also, captions for Figures 6-12 need to explain what the error bars represent.

Reviewer 1 ·

Basic reporting

no comment

Experimental design

no comment

Validity of the findings

no comment

Additional comments

no comment

Reviewer 3 ·

Basic reporting

no comment

Experimental design

no comment

Validity of the findings

no comment

Additional comments

Dear Authors,
Thank you for revising the MS according to the proposed comments and suggestions. Though the authors have addressed most of the comments. Still, I feel there is a scope for further improvement. I believe these minor corrections will help in improving the overall work presentation.

 In the abstract, add a brief line describing the miRNA targeting PDF genes.
 Add “miRNA” in the keywords.
 Supplementary Fig 7 (GO results) and Table S1 (miRNA) are not cited in the text.
 In supplementary Table S1, the vital information is missing. Please add the following information in different columns 1) miRNA name 2) Target gene ID 3) Gene name 4) Expectation 5) miRNA_start 6) miRNA_end 7) Target start 8) Target end 9) miRNA_aligned_fragment 10) alignment 11) Target_aligned_fragment 12) Inhibition. The miRNA result table should comprise of these 12 columns. From the given table, readers cannot identify this vital information of miRNAs.
 Please add a subheading for GO and miRNA results, “GO annotation and miRNA prediction”. Also, add the GO terms ID in the text.
 In the discussion, please support the finding of miRNA targeting PDF genes. Is there any previous report of PDF genes in any organisms targeted by different miRNA families?
 Please give a subheading to the last paragraph. “conclusion”.